# STEP-BY-STEP REASONING FOR MATH PROBLEMS VIA TWISTED SEQUENTIAL MONTE CARLO

**Shengyu Feng**[1]* **Xiang Kong**[2] **Shuang Ma**[2] **Aonan Zhang**[2]
**Dong Yin**[2] **Chong Wang**[2] **Ruoming Pang**[2] **Yiming Yang**[1]
[1]Language Technologies Institute, Carnegie Mellon University [2]Apple

## ABSTRACT

Augmenting the multi-step reasoning abilities of Large Language Models (LLMs) has been a persistent challenge. Recently, verification has shown promise in improving solution consistency by evaluating generated outputs. However, current verification approaches suffer from sampling inefficiencies, requiring a large number of samples to achieve satisfactory performance. Additionally, training an effective verifier often depends on extensive process supervision, which is costly to acquire. In this paper, we address these limitations by introducing a novel verification method based on Twisted Sequential Monte Carlo (TSMC). TSMC sequentially refines its sampling effort to focus exploration on promising candidates, resulting in more efficient generation of high-quality solutions. We apply TSMC to LLMs by estimating the expected future rewards at partial solutions. This approach results in a more straightforward training target that eliminates the need for step-wise human annotations. We empirically demonstrate the advantages of our method across multiple math benchmarks, and also validate our theoretical analysis of both our approach and existing verification methods.

## 1 INTRODUCTION

In recent years, Large Language Models (LLMs) have achieved significant breakthroughs across various domains (Park et al., 2023; Kaddour et al., 2023; Song et al.; Li et al., 2023a; Chen et al., 2023; Zheng et al., 2023; Wang et al., 2024a). However, their performance in multi-step reasoning tasks, such as solving complex mathematical or coding problems, remains notably constrained (Huang et al., 2023; Lightman et al., 2024). A key challenge arises from the high sensitivity of these tasks to individual errors in each reasoning step. Autoregressive LLMs, in particular, struggle to maintain consistency throughout the reasoning process, leading to solutions that are prone to mistakes or logical inconsistencies (Shen et al., 2021; Cobbe et al., 2021).

Verification (Cobbe et al., 2021; Uesato et al., 2022; Lightman et al., 2024) has emerged as an effective strategy to mitigate these issues. In a typical verification process, multiple solutions are sampled from the generator (LLM) and evaluated by an external verifier. The verification results are then used to adjust the weight of each solution in determining the final answer. Since verification is generally simpler than generation, it tends to achieve higher accuracy and consistency compared to methods that rely solely on the generator, such as majority voting (Wang et al., 2023c). There are two primary types of verifiers: *Outcome Reward Model (ORM)* (Cobbe et al., 2021) and *Process Reward Model (PRM)* (Uesato et al., 2022). ORM evaluates the fully generated solution with a single scalar output representing the confidence score, and its training is straightforward: using outcome supervision based on comparing generated answers with ground truth. In contrast, PRM focuses on providing rewards at each step of the reasoning process, giving more detailed feedback on the intermediate steps. Although empirical evidence suggests that PRM outperforms ORM (Lightman et al., 2024), there is no simple metric to evaluate the correctness of each step and efficiently collecting process supervision for such intermediate steps remains a great challenge.

Despite being promising, existing verification methods are still limited in the following two areas:

---

*Work done during internship at Apple. Correspondence to: Shengyu Feng (`shengyuf@cs.cmu.edu`) and Xiang Kong (`xiang_kong@apple.com`).

- **(Problem I) Low sampling efficiency:** Current verification methods only evaluate fully generated solutions, without refining their quality during the generation process. Sampling efforts would be wasted on partial solutions that are clearly incorrect. As a result, a large number of samples are needed to obtain even one correct solution, making the process inefficient and resource-intensive.
- **(Problem II) Difficulty in obtaining process supervision:** Training powerful verifiers like PRM requires detailed step-wise supervision. Existing approaches either rely on human effort (Uesato et al., 2022; Lightman et al., 2024) or tree search (Wang et al., 2023b; Luo et al., 2024) for intermediate step annotations. However, both approaches are inefficient and lack scalability, limiting their practical application for large-scale tasks.

To address these two major problems in existing verification methods, we propose a novel approach based on *Twisted Sequential Monte Carlo (TSMC)* (Doucet et al., 2001; Del Moral et al., 2006; Briers et al., 2009; Chopin & Papaspiliopoulos, 2020). TSMC is a significant advancement in the *Importance Sampling (IS)* technology. Building on the foundation of Sequential Monte Carlo (SMC), TSMC is intended to enhance the sampling efficiency of IS in the high-dimensional space. It employs a series of intermediate target distributions at each resampling step, which are defined through twist functions. This function strategically guides the samples towards the high-density region in the target distribution. By retaining the most promising samples, TSMC effectively reduces the variance in estimated quantities and boosts the efficiency of the sampling process.

Notably, the application of TSMC to improve the verification method in LLMs has not been explored previously, making our study the first of its kind in this area. Our approach is inspired by the realization that existing verification methods employing reward-weighted majority voting (Li et al., 2023b) essentially performs IS, where the sampling efficiency deteriorates as the disparity between the proposal distribution (which generates potential solutions) and the target distribution (concentrated around correct solutions) widens. We identify Problem I—low sampling efficiency—as a consequence of high variance in IS when there is a substantial deviation between the proposal and target distributions. In multi-step reasoning, even minor discrepancies at each step can cumulate into a substantial mismatch between the two distributions. We therefore apply TSMC to improve the sampling efficiency of verification by focusing the sampling effort on promising partial solutions during the intermediate decoding process. We show that the optimal twist function in our case, which is used to guide the sampling of TSMC, is proportional to expected future rewards, also known as the value function. The value function could be simply learned through a neural regressor on the data independently sampled from the generator. This simplifies the training target by eliminating the need for human annotations or tree search. We also highlight the relationship between TSMC and PRM in existing verification methods, allowing for a comprehensive analysis of bias and variance.

We compare our proposed method with baseline approaches on two math benchmarks: GSM8K (Cobbe et al., 2021) and MATH (Hendrycks et al., 2021), utilizing fine-tuned models from Llemma-7B (Azerbayev et al., 2023) and DeepSeek-7B (Shao et al., 2024) as the generators. Our results indicate that TSMC consistently improves both the quality of the generated solutions and the overall verification performance. Additionally, we empirically validate the theoretical advantage of TSMC as an unbiased estimator with reduced variance, further highlighting its effectiveness.

Our main contributions can be summarized as follows:

1. We propose a novel method based on TSMC that enhances the sampling efficiency of verification and reduces the reliance on process supervision, which is usually obtained through human annotations or tree search, in training verifiers.
2. We introduce a new theoretical framework for analyzing verification methods, providing deeper insights into their effectiveness and limitations.
3. Our empirical results demonstrate that TSMC consistently outperforms existing verification methods across multiple math benchmarks, utilizing various generators.

## 2 PRELIMINARIES

### 2.1 LLMs FOR MATH

Following Lightman et al. (2024), we fix the generator without further fine-tuning via reinforcement learning. For a problem statement $\mathbf{x}_0$, a (tokenized) candidate solution can be sampled from the

generator, denoted as $\mathbf{x}_{1:T} \sim p(\cdot|\mathbf{x}_0)$. For simplicity, we always assume the dependence on $\mathbf{x}_0$ and no longer explicitly write it out in the following text. The solution is assumed to be decomposable as $\mathbf{x}_{1:T} = [\mathbf{x}_1, \cdots, \mathbf{x}_T]$, where $\mathbf{x}_i$ is a variable-length reasoning step. By default, the LLM generates all steps in an autoregressive manner, i.e., $\mathbf{x}_t \sim p(\cdot|\mathbf{x}_{1:t-1})$. Each solution $\mathbf{x}_{1:T}$ contains the reasoning process and an answer to the problem, with examples shown in Appendix F. We represent the extracted answer from the solution as $a = \text{Ans}(\mathbf{x}_{1:T})$, and its correctness as $\phi(a)$, which is 1 if it is correct (matched with the ground-truth answer) and 0 otherwise.

The primary methods for solving math problems with LLMs include majority voting (Wang et al., 2023c) and verification (Cobbe et al., 2021; Uesato et al., 2022; Lightman et al., 2024).

**Majority Voting.** Majority voting independently samples $N$ (tokenized) candidate solutions $\{\mathbf{x}_{1:T}^i\}_{i=1}^N$ from the generator. It selects the final answer as the one with the most votes, i.e., $a^* = \arg\max_a \sum_{i=1}^N \mathbb{I}(a_i = a)$, where $\mathbb{I}(\cdot)$ is the indicator function.

**Verification.** Verification introduces an external verifier $r(\cdot)$ to evaluate the $N$ solutions produced by the LLM generator. Existing verifiers can be roughly divided into two kinds: the outcome reward model (ORM) family and the process reward model (PRM) family. ORM directly evaluates the confidence score for each complete solution as $s = r_{ORM}(\mathbf{x}_{1:T})$, while PRM aggregates the confidence scores of sub-sequences as $s = r_{PRM}(\mathbf{x}_{1:T}) = \text{Aggr}(\{r_{PRM}(\mathbf{x}_t|\mathbf{x}_{1:t-1})\}_{t=1}^T)$. Here, $r_{PRM}(\mathbf{x}_t|\mathbf{x}_{1:t-1})$ corresponds to the process reward, and $\text{Aggr}(\cdot)$ is the aggregation function such as the minimum or product:

$$\text{min} = \min\{r_{PRM}(\mathbf{x}_t|\mathbf{x}_{1:t-1})\}_{t=1}^T, \qquad \text{prod} = \prod_{t=1}^T r_{PRM}(\mathbf{x}_t|\mathbf{x}_{1:t-1}). \tag{1}$$

The final answer could either be selected from the solution with the highest score $a^* = \arg\max_{a^i} s^i$ (best-of-$N$), or the answer with the highest total weight $a^* = \arg\max_a \sum_{i=1}^N s^i \mathbb{I}(a_i = a)$ (weighted majority voting) (Li et al., 2023b). In this work, we mainly develop our method on top of the weighted majority voting due to its empirical better performance (Sun et al., 2024).

## 2.2 Importance Sampling and Twisted Sequential Monte Carlo

**Importance Sampling.** Consider a target distribution $\sigma(\mathbf{x}_{1:T}) = \frac{\tilde{\sigma}(\mathbf{x}_{1:T})}{Z^\sigma}$, where $\tilde{\sigma}(\mathbf{x}_{1:T}) \geq 0$ is the unnormalized probability density and $Z^\sigma = \int_{\mathbf{x}_{1:T}} \tilde{\sigma}(\mathbf{x}_{1:T})d\mathbf{x}_{1:T}$ is the normalizing factor, typically intractable. For a given function $h(\mathbf{x}_{1:T})$, it could be difficult to estimate its expectation under $\sigma(\mathbf{x}_{1:T})$ via direct sampling. Importance sampling (IS) (Robert & Casella, 2000) instead introduces a proposal distribution $q(\mathbf{x}_{1:T})$ and provides an estimator of the expectation as

$$\mathbb{E}_{\sigma(\mathbf{x}_{1:T})}[h(\mathbf{x}_{1:T})] = \frac{1}{Z^\sigma}\mathbb{E}_{q(\mathbf{x}_{1:T})}[\frac{\tilde{\sigma}(\mathbf{x}_{1:T})}{q(\mathbf{x}_{1:T})}h(\mathbf{x}_{1:T})] = \frac{\mathbb{E}_{q(\mathbf{x}_{1:T})}[\frac{\tilde{\sigma}(\mathbf{x}_{1:T})}{q(\mathbf{x}_{1:T})}h(\mathbf{x}_{1:T})]}{\mathbb{E}_{q(\mathbf{x}_{1:T})}[\frac{\tilde{\sigma}(\mathbf{x}_{1:T})}{q(\mathbf{x}_{1:T})}]}. \tag{2}$$

Here, $\frac{\tilde{\sigma}(\mathbf{x}_{1:T})}{q(\mathbf{x}_{1:T})}$ is known as the *importance weight* $w(\mathbf{x}_{1:T})$. Using some $q(\mathbf{x}_{1:T})$ that is easy to sample from, we can leverage Equation 2 to estimate the expectation via the Monte Carlo method:

$$\mathbb{E}_{\sigma(\mathbf{x}_{1:T})}[h(\mathbf{x}_{1:T})] \approx \sum_{i=1}^N \frac{w(\mathbf{x}_{1:T}^i)}{\sum_{j=1}^N w(\mathbf{x}_{1:T}^j)}h(\mathbf{x}_{1:T}^i), \qquad \mathbf{x}_{1:T}^i \sim q(\mathbf{x}_{1:T}). \tag{3}$$

Although ideally a zero variance of the importance weight could be achieved when $q(\mathbf{x}_{1:T}) = \sigma(\mathbf{x}_{1:T})$, such a case rarely holds in practice. Remarkably, the distribution mismatches at each step are accumulated as the generation proceeds, leading to an exponentially increasing variance with respect to $T$ (Doucet & Johansen, 2009). Such a limitation makes IS inefficient in the high-dimensional space, since extensive sampling is needed to reduce the variance.

**Twisted Sequential Monte Carlo.** Twisted Sequential Monte Carlo (TSMC) enhances the sampling efficiency of IS by modifying the marginal distribution of the proposal, $q(\mathbf{x}_{1:t})$, to a more informative intermediate distribution, $\pi_t(\mathbf{x}_{1:t})$. The aim is to ensure that partial sequences from $\pi_t(\mathbf{x}_{1:t})$ are more likely to produce high-density samples in the final target distribution $\sigma(\mathbf{x}_{1:T})$.

Here, $\{\pi_t\}_{t=1}^T$ are known as the (twisted) intermediate targets, where $\pi_t(\mathbf{x}_{1:t}) = \frac{\tilde{\pi}_t(\mathbf{x}_{1:t})}{Z_t^\pi}$, and the final target is aligned with $\tilde{\pi}_T \equiv \tilde{\sigma}$. In standard Sequential Monte Carlo, $\pi_t(\mathbf{x}_{1:t})$ is typically the target marginal $\sigma(\mathbf{x}_{1:t}) = \sum_{\mathbf{x}_{t+1:T}} \sigma(\mathbf{x}_{1:T})$, to ensure that at each time step, the marginal distribution matches the target. However, if our primary interest is only the final target $\sigma(\mathbf{x}_{1:T})$, we are free to design $\{\pi_t\}_{t=1}^{T-1}$ on the specific problem at hand, leading to the flexibility of the TSMC method.

TSMC operates recursively, alternating between generation and resampling. At each step, TSMC takes the input of $N$ partial sequences, $\{\mathbf{x}_{1:t-1}^i\}_{i=1}^N$, following the distribution $\pi_{t-1}(\mathbf{x}_{1:t-1})$, and extends these sequences by sampling the next step from the proposal, i.e., $\mathbf{x}_t \sim q(\cdot|\mathbf{x}_{1:t-1})$. It computes the *incremental importance weight* for each sequence as

$$w_t(\mathbf{x}_{1:t}) = \frac{\tilde{\pi}_t(\mathbf{x}_{1:t})}{q(\mathbf{x}_t|\mathbf{x}_{1:t-1})\tilde{\pi}_{t-1}(\mathbf{x}_{1:t-1})}. \tag{4}$$

These weights are used to approximate the distribution $\pi_t$ by resampling the partial sequences from a categorical distribution with the self-normalized weights:

$$\mathbf{x}_{1:t}^i \leftarrow \mathbf{x}_{1:t}^{\omega_i}, \qquad \omega^i \sim \texttt{Cat}(\{\frac{w_t(\mathbf{x}_{1:t}^i)}{\sum_{j=1}^N w_t(\mathbf{x}_{1:t}^j)}\}_{i=1}^N), \qquad i = 1, \cdots, N. \tag{5}$$

This new set of $N$ sequences would serve as the input to the next step of TSMC. With informative intermediate targets, the resampling step could promptly discard the sequences with a low potential in the target distribution and avoid a large variance in the importance weights. More importantly, since $\pi_T(\mathbf{x}_{1:T})$ is matched with the target $\sigma(\mathbf{x}_{1:T})$, TSMC always yields an unbiased estimator of $\sum_{\mathbf{x}_{1:T}} \tilde{\sigma}(\mathbf{x}_{1:T})h(\mathbf{x}_{1:T})$ regardless of the twist functions (Del Moral, 2004).

## 3 METHODOLOGY

### 3.1 EXISTING VERIFICATION METHODS ARE PERFORMING IMPORTANCE SAMPLING

The motivation of our method is based on the observation that existing verification methods are essentially performing IS. To see this, compare the normalized voting weight of each answer in majority voting and weighted majority voting when $N$ is infinitely large:

$$\lim_{N\to\infty} \frac{\sum_{i=1}^N \mathbb{I}(a_i = a)}{N} = \mathbb{E}_{p(\mathbf{x}_{1:T})}[\mathbb{I}(\texttt{Ans}(\mathbf{x}_{1:T}) = a)] \qquad \text{(majority voting)} \quad (6)$$

$$\lim_{N\to\infty} \frac{\sum_{i=1}^N s_i\mathbb{I}(a_i = a)}{N} = \mathbb{E}_{p(\mathbf{x}_{1:T})}[r(\mathbf{x}_{1:T})\mathbb{I}(\texttt{Ans}(\mathbf{x}_{1:T}) = a)] \quad \text{(weighted majority voting)} \quad (7)$$

It can be seen that the weighting process actually introduces a factor $r(\mathbf{x}_{1:T})$ with a similar role of the importance weight in Equation 2. In particular, we can let $\tilde{\sigma}(\mathbf{x}_{1:T}) = p(\mathbf{x}_{1:T})r(\mathbf{x}_{1:T})$ and treat weighted majority voting as IS to estimate the answer voting weight

$$w(a) = \sum_{\mathbf{x}_{1:T}} \tilde{\sigma}(\mathbf{x}_{1:T})\mathbb{I}(\texttt{Ans}(\mathbf{x}_{1:T}) = a). \tag{8}$$

However, as described in Section 2.2, the importance weight in IS suffers from a large variance in the high-dimensional space, so do the estimation objectives according to Proposition 3.1.

**Proposition 3.1.** *For IS with the target $\sigma(\mathbf{x}_{1:T})$ and proposal $q(\mathbf{x}_{1:T})$, up to a constant $C$ independent of $q(\mathbf{x}_{1:T})$, the following identity in the variance holds for the set of all answers $\mathcal{A}$:*

$$\sum_{a\in\mathcal{A}} \mathbb{V}_q[\frac{\tilde{\sigma}(\mathbf{x}_{1:T})\mathbb{I}(\texttt{Ans}(\mathbf{x}_{1:T}) = a)}{q(\mathbf{x}_{1:T})}] = \mathbb{V}_q[\frac{\tilde{\sigma}(\mathbf{x}_{1:T})}{q(\mathbf{x}_{1:T})}] + C. \tag{9}$$

We include the proof in Appendix A.1. This issue also accounts for problem I, that is, many samples are needed to reduce the variance of the estimator. Therefore, we aim to address this problem via TSMC, which provides the unbiased estimator of $w(a)$, but with less variance. We visualize the comparison between existing IS-based verification and our TSMC-based verification in Figure 1.

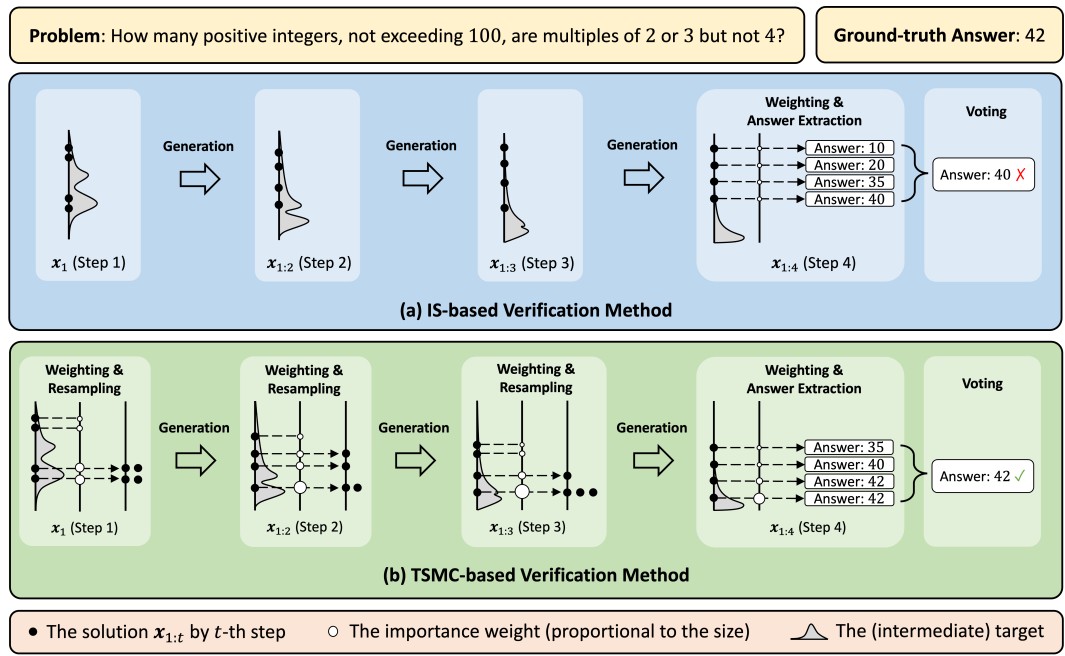

Figure 1: IS-based verification vs. TSMC-based verification. (a) **Typical IS-based verification** only weights (verifies) the solutions until they are fully generated, which often leads to generating incorrect solutions with high probability, aka low sampling efficiency. (b) **Our TSMC-based verification** weights and resamples partial solutions at each step of the generation process. This sequential resampling process reduces the discrepancy between the proposal and target distributions, improving the overall correctness of the generated solutions and thus the sampling efficiency.

## 3.2 VERIFICATION VIA TSMC

The optimal reward model in our problem is simply the correctness function of each solution, i.e., $r^*(\mathbf{x}_{1:T}) = \phi(\texttt{Ans}(\mathbf{x}_{1:T}))$. In the following section, we fix our target distribution as

$$\sigma(\mathbf{x}_{1:T}) = \frac{p(\mathbf{x}_{1:T})\phi(\texttt{Ans}(\mathbf{x}_{1:T}))}{Z^{\sigma}}, \tag{10}$$

since it corresponds to the actual target distribution we try to sample from.

To utilize TSMC for verification, we need to decide the proposal distribution $q(\mathbf{x}_t|\mathbf{x}_{1:t-1})$ and intermediate targets $\{\pi_t\}_{t=1}^{T-1}$. Following Zhao et al. (2024), we define the intermediate targets through the nonnegative twist functions $\{\psi_t\}_{t=1}^{T-1}$, which are designed to be optimized:

$$\pi_t(\mathbf{x}_{1:t}) = \frac{p(\mathbf{x}_{1:t})\psi_t(\mathbf{x}_{1:t})}{Z_t^{\pi}}. \tag{11}$$

Let $\psi_0(\mathbf{x}_0) \equiv 1$ and $\psi_T(\mathbf{x}_{1:T}) \equiv \phi(\texttt{Ans}(\mathbf{x}_{1:T}))$, the incremental importance weight is given by

$$w_t(\mathbf{x}_{1:t}) = \frac{p(\mathbf{x}_t|\mathbf{x}_{1:t-1})\psi_t(\mathbf{x}_{1:t})}{q(\mathbf{x}_t|\mathbf{x}_{1:t-1})\psi_{t-1}(\mathbf{x}_{1:t-1})}. \tag{12}$$

Zhao et al. (2024) have shown that the optimal proposal and intermediate targets correspond to

$$q_t^*(\mathbf{x}_t|\mathbf{x}_{1:t-1}) = \frac{\sigma(\mathbf{x}_{1:t})}{\sigma(\mathbf{x}_{1:t-1})} \quad \text{and} \quad \pi_t^*(\mathbf{x}_{1:t}) = \sigma(\mathbf{x}_{1:t}). \tag{13}$$

However, it is hard to directly apply these optimal choices in our case. We outline the reasons and our approach as follows.

**Proposal.** There are two challenges that prevent us from using the optimal proposal. First, the combinatorial nature of the step $\mathbf{x}_t$, which consists of multiple tokens, makes $q_t^*(\mathbf{x}_t|\mathbf{x}_{1:t-1})$ generally intractable. Moreover, even if we can approximately sample from it, we still encounter the second challenge, the large variance in $q_t^*(\mathbf{x}_t|\mathbf{x}_{1:t-1})$ caused by the high dimensionality of $\mathbf{x}_t$. This would result in the *weight degeneracy* issue (Naesseth et al., 2019) of TSMC, where the incremental importance weights would be dominated by a single sample, resulting in a poor diversity of solutions after resampling. We therefore simply let $q(\mathbf{x}_t|\mathbf{x}_{1:t-1}) = p(\mathbf{x}_t|\mathbf{x}_{1:t-1})$ for ease of sampling, and weight degeneracy would be alleviated when $q(\mathbf{x}_t|\mathbf{x}_{1:t-1})$ cancels in Equation 12.

**Intermediate targets.** The intermediate targets in Equation 13 are no longer optimal when we use $p(\mathbf{x}_t|\mathbf{x}_{1:t-1})$ as our proposal. However, it is also hard to solve the globally optimal intermediate targets for an arbitrary proposal. We instead seek to sequentially derive the locally optimal twists in a greedy manner. Since our ultimate goal is to estimate the answer weights $w(a)$, we start by looking for the optimal intermediate target $\pi_{T-1}^*(\mathbf{x}_{1:T-1})$ to minimize the variance of the incremental importance weight in the last TSMC step. We prove the following proposition in Appendix A.2.

**Proposition 3.2.** *Given an intermediate target $\pi_t(\mathbf{x}_{1:t})$ and the proposal $q(\mathbf{x}_t|\mathbf{x}_{1:t-1})$, the optimal $\pi_{t-1}(\mathbf{x}_{1:t-1})$ in minimizing the variance of the incremental importance weight corresponds to*

$$\pi_{t-1}^q(\mathbf{x}_{1:t-1}) \propto \sqrt{\sum_{\mathbf{x}_t} \frac{\pi_t(\mathbf{x}_{1:t})^2}{q(\mathbf{x}_t|\mathbf{x}_{1:t-1})}}. \tag{14}$$

Taking $t = T$ and $q = p$ implies $\pi_{T-1}^p(\mathbf{x}_{1:T-1}) \propto p(\mathbf{x}_{1:T-1})\sqrt{\sum_{\mathbf{x}_T} p(\mathbf{x}_T|\mathbf{x}_{1:T-1})\phi(\text{Ans}(\mathbf{x}_{1:T}))}$. Here we apply the fact that $\phi(\text{Ans}(\mathbf{x}_{1:T}))^2 = \phi(\text{Ans}(\mathbf{x}_{1:T}))$ as it is binary. If we fix the intermediate target as the choice above, we could further propagate the derivation to previous steps by recursively applying Proposition 3.2, getting the locally optimal intermediate targets for $t < T$ as

$$\pi_t^p(\mathbf{x}_{1:t}) \propto p(\mathbf{x}_{1:t})\sqrt{\sum_{\mathbf{x}_{t+1:T}} p(\mathbf{x}_{t+1:T}|\mathbf{x}_{1:t})\phi(\text{Ans}(\mathbf{x}_{1:T}))}. \tag{15}$$

In particular, $\sum_{\mathbf{x}_{t+1:T}} p(\mathbf{x}_{t+1:T}|\mathbf{x}_{1:t})\phi(\text{Ans}(\mathbf{x}_{1:T}))$ actually represents the value function $V^p(\mathbf{x}_{1:t})$ in reinforcement learning (Ouyang et al., 2022). Hence, the locally optimal twists are given by

$$\psi_t^p(\mathbf{x}_{1:t}) \propto \sqrt{V^p(\mathbf{x}_{1:t})}. \tag{16}$$

## 3.3 CONNECTION WITH THE PRM

Based on our above choices, the incremental importance weights in Equation 12 becomes

$$w_t^p(\mathbf{x}_{1:t}) = \frac{\psi_t^p(\mathbf{x}_{1:t})}{\psi_{t-1}^p(\mathbf{x}_{1:t-1})} \propto \sqrt{\frac{V^p(\mathbf{x}_{1:t})}{V^p(\mathbf{x}_{1:t-1})}}. \tag{17}$$

The incremental importance weight could also be treated as a measurement of the step quality, similar to the process reward in PRMs. To further augment this connection, note that

$$\prod_{t=1}^T w^p(\mathbf{x}_{1:t}) = \prod_{t=1}^T \frac{\psi_t^p(\mathbf{x}_{1:t})}{\psi_{t-1}^p(\mathbf{x}_{1:t-1})} = \frac{\psi_T^p(\mathbf{x}_{1:T})}{\psi_0^p(\mathbf{x}_0)} = \phi(\text{Ans}(\mathbf{x}_{1:T})), \tag{18}$$

which is in the same format as the PRM with `prod` aggregation. The key observation here is that TSMC always yields an unbiased estimator of the importance weight $\phi(\text{Ans}(\mathbf{x}_{1:T}))$, when there is no estimation error of $V^p$. We continue to compare this estimator to some existing PRMs.

**The PRM learned through automatic supervision.** This class of PRMs (Wang et al., 2023a; Luo et al., 2024) computes the process reward by evaluating the value function at each partial solution with respect to a roll-out policy $\mu$. The solution confidence score will be computed as

$$r_{PRM}(\mathbf{x}_{1:T}) = \text{Aggr}(\{r(\mathbf{x}_t|\mathbf{x}_{1:t-1})\}_{t=1}^T) = \text{Aggr}(\{V^\mu(\mathbf{x}_{1:t})\}_{t=1}^T). \tag{19}$$

However, such an estimator is always biased no matter `min` or `prod` is used for aggregation.

**The PRM learned through human supervision.** The human supervision is generated through the logical sense of step correctness. We formally establish the definition of step correctness in Definition A.1 and prove the following proposition in Appendix A.3.

**Proposition 3.3.** *The PRM corresponding to the step correctness can be expressed as*

$$r_{PRM}(\mathbf{x}_t|\mathbf{x}_{1:t-1}) = \mathbb{I}(\sigma(\mathbf{x}_{1:t}) > 0). \tag{20}$$

Therefore, the solution confidence score from this PRM is always an unbiased estimator of $\phi(\text{Ans}(\mathbf{x}_{1:T}))$ for both min and prod aggregation. However, using $\mathbb{I}(\sigma(\mathbf{x}_{1:t}) > 0)$ for the intermediate target, as tried by Uesato et al. (2022), does not effectively reduce the sampling variance or improve verification performance since it ignores the likelihood of the proposal $p(\mathbf{x}_{t+1:T}|\mathbf{x}_{1:t})$.

## 3.4 VALUE FUNCTION ESTIMATION

The approximation of $\{\psi_t^p\}_{t=1}^{T-1}$ and $r^*$ can be consolidated into a single learning task: estimating the value function $V^p$. We therefore use a single neural model parameterized by $\theta$ for the approximation. Estimating the value function through independently sampled data from a fixed policy (generator) is a well-studied topic (Bertsekas, 2012). It therefore eliminates the need for explicit process supervision during training, as outlined in Problem II.

In this paper, we adopt the Contrastive Twist Learning (CTL) method developed by Zhao et al. (2024). It is hard to directly approximate $\pi_t^p$ via Monte Carlo sampling, so we instead approximate the target marginal $\sigma(\mathbf{x}_{1:t})$ to estimate the value function, and take the square root of the estimated value function during the inference time. Let $V^\theta$ be our estimated value function of $V^p$, and define the intermediate target $\pi_t^\theta(\mathbf{x}_{1:t}) = \frac{p(\mathbf{x}_{1:t})V^\theta(\mathbf{x}_{1:t})}{Z_t^\pi(\mathbf{x}_{1:t})}$. CTL minimizes the KL divergence between the target marginal distributions and the intermediate targets:

$$\min_\theta L_{CTL}(\theta) = \min_\theta \sum_{t=1}^{T} D_{KL}(\sigma(\mathbf{x}_{1:t})\|\pi_t^\theta(\mathbf{x}_{1:t})), \tag{21}$$

whose gradient at $t$-th step can be derived as

$$\mathbb{E}_{\sigma(\mathbf{x}_{1:t})}[\nabla_\theta \log V^\theta(\mathbf{x}_{1:t})] - \mathbb{E}_{\pi_t^\theta(\mathbf{x}_{1:t})}[\nabla_\theta \log V^\theta(\mathbf{x}_{1:t})]. \tag{22}$$

We approximate the gradient in the first term via rejection sampling, and the gradient in the second term via importance sampling, as done in Equation 3. We include more training details in Appendix C.2 and summarize our entire TSMC-based verification algorithm in Appendix B.

## 4 EXPERIMENTS

### 4.1 EXPERIMENTAL SETUP

We briefly outline our experimental setup in this section and include more details in Appendix C.

**Datasets.** Building on prior work (Uesato et al., 2022; Lightman et al., 2024; Wang et al., 2023a), we assess our TSMC method using two widely used math datasets: GSM8K (Cobbe et al., 2021) and MATH (Hendrycks et al., 2021). For GSM8K, we evaluate model performance on all testing instances. While for MATH, we follow Lightman et al. (2024) to select a representative subset of 500 testing instances, referred to as MATH500 in the following text.

**Generators.** We fine-tune two solution generators from pretrained Llemma-7B (Azerbayev et al., 2023) and DeepSeek-7B (Shao et al., 2024), respectively. Following Sun et al. (2024), we use the filtered PRM800K (Lightman et al., 2024) as the supervised fine-tuning dataset.

**Baselines.** We compare our method to both non-verification methods, including zero-shot greedy decoding and majority voting (MV) (Wang et al., 2023c), and verification methods based on weighted majority voting (WMV). For verification methods, we use various types of verifiers, including the ORM, the PRM trained with human supervision on PRM800K (Lightman et al., 2024)),

and the PRM trained with automatic supervision on MATH-SHEPHERD (Wang et al., 2023a). We employ `min` for aggregation on both PRMs. For each generator, we use the same pretrained model to train the reward models and value function, with a linear head applied. The problem solving rate (in %) is used as the comparison metric.

**TSMC details.** Our TSMC is applied at the step level. We implement a warm-up stage that skips resampling in the initial stage, setting this threshold at 50 tokens across all experiments. A maximum of five resampling steps is allowed to reduce the latency. For sequences that terminate early, we assign an incremental importance weight of 1 during the remaining resampling steps. We employ stratified sampling (Kitagawa, 1996) for resampling to reduce variance. Due to the resource constraint, we perform resampling over a mini-batch of $M$ samples instead of the full batch of $N$ solutions. The batch set is fixed as $M = 40$ by default in the following experiments.

## 4.2 MAIN RESULTS

To verify whether TSMC really improves the sampling efficiency with better solution quality, we also perform the majority voting on the solutions generated by TSMC. We denote this method as TSMC + MV, and the complete TSMC as TSMC + WMV. We present our main results in Table 1.

Table 1: Comparative results in the problem solving rate (%) on GSM8K and MATH500 datasets. We use two generators fine-tuned from pretrained Llemma-7B and DeepSeek-7B, respectively. We bold the best results in each category. The voting is performed on 240 samples.

| Generators | Methods | GSM8K | MATH500 |
|---|---|---|---|
| Llemma-7B | Greedy | 38.2 | 19.4 |
| | MV | 72.5 | 41.2 |
| | WMV w. ORM | 78.7 | 43.0 |
| | WMV w. PRM (PRM800K) | 73.6 | 43.2 |
| | WMV w. PRM (SHEPHERD) | 79.2 | 43.6 |
| | TSMC + MV (Ours) | 78.1 | 44.2 |
| | TSMC + WMV (Ours) | **80.4** | **46.4** |
| DeepSeek-7B | Greedy | 61.2 | 30.8 |
| | MV | 86.4 | 52.8 |
| | WMV w. ORM | 86.6 | 55.0 |
| | WMV w. PRM (PRM800K) | 87.0 | 55.2 |
| | WMV w. PRM (SHEPHERD) | 89.5 | 52.6 |
| | TSMC + MV (Ours) | 89.5 | 55.6 |
| | TSMC + WMV (Ours) | **91.7** | **60.8** |

It is evident that TSMC + MV demonstrates a significant improvement over vanilla MV, highlighting its effectiveness in enhancing the overall solution quality. Moreover, TSMC consistently outperforms other methods in terms of final verification performance. It is worth noting that the final verification step in TSMC operates independently of the generator, meaning that a better reward model could further improve TSMC's performance in WMV. Overall, our TSMC-based verification method shows a clear advantage over existing verification methods, with a simpler training target.

## 4.3 IMPACT OF BIAS ON ESTIMATORS

TSMC is characterized by its unbiased estimation of the importance weight, which is $\phi(\text{Ans}(\mathbf{x}_{1:T}))$ in our task. However, since the training error is unavoidable in practice, it remains unclear whether such an unbiased estimation in the theoretical optimal case is meaningful. We therefore examine this problem by comparing different biased and unbiased estimators analyzed in Section 3.3.

We consider both the PRM predicting step correctness, i.e., *PRM (PRM800K)*, and the PRM estimating the value function, including *PRM (SHEPHERD)* and the estimated value function in TSMC, abbreviated as *Value (TSMC)*. PRM (PRM800K) and PRM (SHEPHERD) are applied to independently generated solutions, while Value (TSMC) is applied to TSMC-generated solutions, treating

estimated values as process rewards. Beyond `min` and `prod`, we also consider aggregating the process rewards by taking last-step reward only, yielding an ORM-like unbiased estimator:

$$\texttt{last} = r_{PRM}(\mathbf{x}_T | \mathbf{x}_{1:T-1}). \tag{23}$$

Exceptionally, since the solutions in TSMC have already gone through the resampling process, we use the last incremental weight for `last` in this scenario, recovering the original TSMC process as an unbiased estimator. We compare all estimators in Figure 2, under different settings.

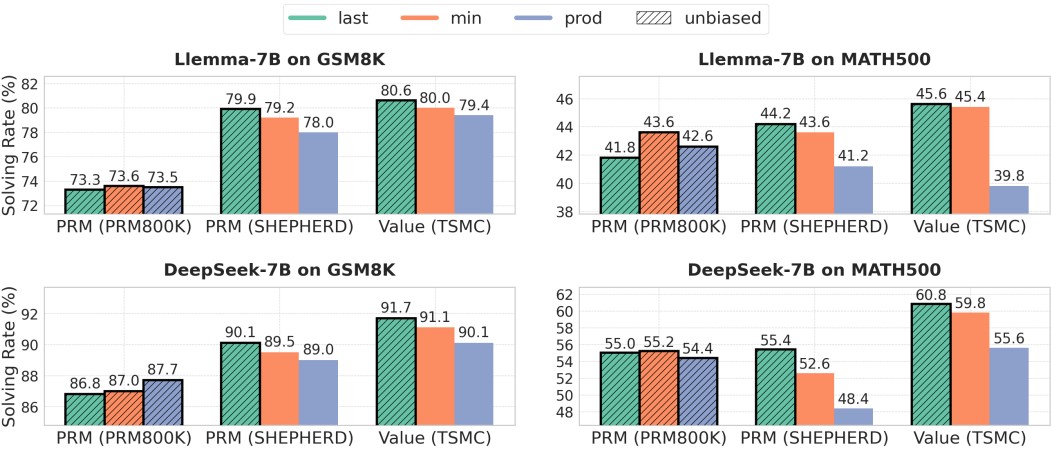

Figure 2: Comparison among all biased and unbiased estimators of the importance weight.

The trend is highly consistent when the estimated value is used as the process reward, i.e., in PRM (SHEPHERD) and Value (TSMC). In this case, `prod` consistently exhibits poor performance across all settings. Since the value at each step lies in $[0, 1]$, the product of values is highly biased towards the solutions with fewer steps. In contrast, `min` could overcome such a bias as its value is insensitive to the number of steps. Its clear advantage against `prod` is also in line with the choice of (Wang et al., 2023a). However, the performance is still consistently worse than the unbiased estimator using `last`. A different pattern shows up in PRM (PRM800K), where the estimated step correctness is used as the process reward. The `min` strategy still achieves the overall best result, but `prod` is also comparably good. We find no advantage of `last` in this case, as all estimators are unbiased. Instead, `prod` and `min` would benefit from ensemble modeling by aggregating multiple scores.

We find our results consistent with the observation from Sun et al. (2024). Basically, the advantage of PRM against ORM is enduring only when both are unbiased estimators of $\phi(\texttt{Ans}(\mathbf{x}_{1:T}))$. When PRM is biased, there is no clear guarantee of better performance against ORM, which is always an unbiased estimator. TSMC instead assimilates the strength of the unbiased estimation from ORM and the intermediate-step modeling from PRM, leading to the best performance.

### 4.4 IMPACT OF VARIANCE ON ESTIMATORS

Besides of being unbiased, TSMC reduces the variance of importance weight through informative twists. To investigate the impact of variance, we consider the following variants of TSMC: using the step correctness predicted by PRM (PRM800K) as the incremental importance weight; using the process reward in PRM (SHEPHERD) as the estimated value function; and using $V^p(\mathbf{x}_{1:t})$ rather than $\sqrt{V^p(\mathbf{x}_{1:t})}$ for the twist, which approximates the target marginal $\sigma(\mathbf{x}_{1:t})$ as the intermediate target. Taking Llemma-7B on MATH500 as an example, we compare their performance in Figure 3.

As noted in Uesato et al. (2022), the use of step correctness to guide intermediate decoding does not enhance performance because it disregards the likelihood from the generator. In contrast, PRM (SHEPHERD) offers a more informative guidance through its estimated value function. Nonetheless, this value is assessed using a different generator $\mu$, resulting in inferior performance compared to employing the approximated $V^p$ as the twist. Finally, utilizing $V^p(\mathbf{x}_{1:t})$ as the twist yields poorer results than using $\sqrt{V^p(\mathbf{x}_{1:t})}$, highlighting the importance of optimizing variance.

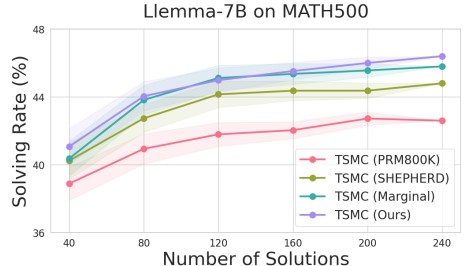 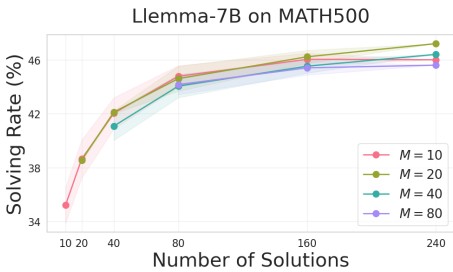

Figure 3: TSMC with different intermediate targets. Variance are visualized across many subsamples of the 240 solutions per problem.

Figure 4: Ablation study on the TSMC batch size. Variance are visualized across many subsamples of the 240 solutions per problem.

## 4.5 SENSITIVITY ANALYSIS OF MODEL PERFORMANCE TO TSMC BATCH SIZE

Different from standard decoding, TSMC is a batch decoding method whose performance is dependent on the batch size $M$. Notably, when $M = 1$, TSMC is reduced to standard decoding. To explore how sensitive TSMC is to its batch size, we vary $M$ over $\{10, 20, 40, 80\}$ and visualize the comparative results in Figure 4. As illustrated, the overall variance stays within a reasonable range, and all outcomes significantly outperform the baselines in Table 1. Consequently, we claim that TSMC is robust to changes in its batch size, provided that the batch size is not too small.

## 5 RELATED WORK

**Verification for reasoning.** Verification has proven to be an effective approach in enhancing the multi-step reasoning ability of LLMs. Two widely adopted verification methods are Outcome Reward Model (ORM) (Cobbe et al., 2021) and Process Reward Model (PRM) (Uesato et al., 2022). Although empirical evidence suggests that PRM outperforms ORM (Lightman et al., 2024), training PRM presents a significant challenge due to the need for process supervision, which is often difficult to obtain. Therefore, recent research has increasingly focused on automatic supervision to train PRMs more efficiently (Wang et al., 2023b; Luo et al., 2024; Wang et al., 2024b).

**(Twisted) Sequential Monte Carlo.** Sequential Monte Carlo (SMC) is a generic statistical inference approach that has been widely applied in various domains, including signal processing (Doucet & Johansen, 2009; Simon J Godsill & West, 2004), financial econometrics (Johannes & Polson, 2010; Creal, 2012), and robotics (Montemerlo et al., 2002; Bailey & Durrant-Whyte, 2006; Thrun et al., 2005). Recently, SMC has been integrated with neural networks to enhance sequential generative models, such as diffusion models (Trippe et al., 2023; Wu et al., 2023) and LLMs (Lew et al., 2023; Zhao et al., 2024). The most relevant work to ours is Zhao et al. (2024), which presents a general framework for controlled text generation using Twisted Sequential Monte Carlo (TSMC). However, our work primarily focuses on multi-step reasoning, under the assumption of a weak generator and long reasoning process. Furthermore, we are the first to bridge TSMC with the predominant verification methods, offering a novel theoretical perspective for interpretability.

## 6 CONCLUSION & LIMITATION

In this paper, we introduce a novel verification method for multi-step reasoning using Twisted Sequential Monte Carlo (TSMC). Our approach sequentially approximates intermediate targets, enhancing the reasoning process of large language models, and improving both solution quality and sampling efficiency. By incorporating step-wise guidance that is learned without human supervision, our method provides a scalable framework for various multi-step reasoning tasks.

Although promising, our method could potentially suffer from the broken parallelism caused by the variable step length, leading to additional inference latency. One possible solution involves blockwise resampling over steps with a fixed number of tokens. Future work could also explore the impact of TSMC batch size and refine algorithmic design for further efficiency gains.

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

# A  PROOFS

Here we include the proofs for all the propositions present in the main paper.

## A.1  PROOF FOR ANSWER WEIGHT VARIANCE

**Proposition 3.1.** *For IS with the target $\sigma(\mathbf{x}_{1:T})$ and proposal $q(\mathbf{x}_{1:T})$, up to a constant $C$ independent of $q(\mathbf{x}_{1:T})$, the following identity in the variance holds for the set of all answers $\mathcal{A}$:*

$$\sum_{a \in \mathcal{A}} \mathbb{V}_q[\frac{\tilde{\sigma}(\mathbf{x}_{1:T})\mathbb{I}(Ans(\mathbf{x}_{1:T}) = a)}{q(\mathbf{x}_{1:T})}] = \mathbb{V}_q[\frac{\tilde{\sigma}(\mathbf{x}_{1:T})}{q(\mathbf{x}_{1:T})}] + C. \tag{9}$$

*Proof.* For simplicity, denote $f_a(\mathbf{x}_{1:T}) = \mathbb{I}(Ans(\mathbf{x}_{1:T}) = a)$. Using the fact that $f_a(\mathbf{x}_{1:T})^2 = f_a(\mathbf{x}_{1:T})$ and $\sum_{a \in \mathcal{A}} f_a(\mathbf{x}_{1:T}) = 1$, we have

$$\sum_{a \in \mathcal{A}} \mathbb{V}_q[\frac{\tilde{\sigma}(\mathbf{x}_{1:T})f_a(\mathbf{x}_{1:T})}{q(\mathbf{x}_{1:T})}] = \sum_{a \in \mathcal{A}}(\mathbb{E}_q[(\frac{\tilde{\sigma}(\mathbf{x}_{1:T})f_a(\mathbf{x}_{1:T})}{q(\mathbf{x}_{1:T})})^2] - \mathbb{E}_q[\frac{\tilde{\sigma}(\mathbf{x}_{1:T})f_a(\mathbf{x}_{1:T})}{q(\mathbf{x}_{1:T})}]^2)$$

$$= \sum_{a \in \mathcal{A}}(\sum_{\mathbf{x}_{1:T}} \frac{\tilde{\sigma}(\mathbf{x}_{1:T})^2 f_a(\mathbf{x}_{1:T})^2}{q(\mathbf{x}_{1:T})} - (\sum_{\mathbf{x}_{1:T}} \tilde{\sigma}(\mathbf{x}_{1:T})f_a(\mathbf{x}_{1:T}))^2)$$

$$= \sum_{\mathbf{x}_{1:T}} \frac{\tilde{\sigma}(\mathbf{x}_{1:T})^2 \sum_a f_a(\mathbf{x}_{1:T})^2}{q(\mathbf{x}_{1:T})} - \sum_{a \in \mathcal{A}}(\sum_{\mathbf{x}_{1:T}} \tilde{\sigma}(\mathbf{x}_{1:T})f_a(\mathbf{x}_{1:T}))^2 \tag{24}$$

$$= \sum_{\mathbf{x}_{1:T}} \frac{\tilde{\sigma}(\mathbf{x}_{1:T})^2}{q(\mathbf{x}_{1:T})} - \sum_{a \in \mathcal{A}}(\sum_{\mathbf{x}_{1:T}} \tilde{\sigma}(\mathbf{x}_{1:T})f_a(\mathbf{x}_{1:T}))^2$$

$$= \mathbb{V}_q[\frac{\tilde{\sigma}(\mathbf{x}_{1:T})}{q(\mathbf{x}_{1:T})}] + C.$$

Here, $C = (\sum_{\mathbf{x}_{1:T}} \tilde{\sigma}(\mathbf{x}_{1:T}))^2 - \sum_{a \in \mathcal{A}}(\sum_{\mathbf{x}_{1:T}} \tilde{\sigma}(\mathbf{x}_{1:T})f_a(\mathbf{x}_{1:T}))^2$ is independent of $q(\mathbf{x}_{1:T})$. □

## A.2  PROOF FOR LOCALLY OPTIMAL INTERMEDIATE TARGETS

**Proposition 3.2.** *Given an intermediate target $\pi_t(\mathbf{x}_{1:t})$ and the proposal $q(\mathbf{x}_t|\mathbf{x}_{1:t-1})$, the optimal $\pi_{t-1}(\mathbf{x}_{1:t-1})$ in minimizing the variance of the incremental importance weight corresponds to*

$$\pi_{t-1}^q(\mathbf{x}_{1:t-1}) \propto \sqrt{\sum_{\mathbf{x}_t} \frac{\pi_t(\mathbf{x}_{1:t})^2}{q(\mathbf{x}_t|\mathbf{x}_{1:t-1})}}. \tag{14}$$

*Proof.* Note that we have the expectation of importance weight

$$\mathbb{E}_{p,\pi_{t-1}}[\frac{\pi_t(\mathbf{x}_{1:t})}{p(\mathbf{x}_t|\mathbf{x}_{1:t-1})\pi_{t-1}(\mathbf{x}_{1:t-1})}] = 1. \tag{25}$$

Using the fact that $\mathbb{V}[X] = \mathbb{E}[X^2] - \mathbb{E}[X]^2$, we write the variance of the importance weight as

$$\mathbb{V}_{p,\pi_{t-1}}[\frac{\pi_t(\mathbf{x}_{1:t})}{q(\mathbf{x}_t|\mathbf{x}_{1:t-1})\pi_{t-1}(\mathbf{x}_{1:t-1})}] = \mathbb{E}_{q,\pi_{t-1}}[(\frac{\pi_t(\mathbf{x}_{1:t})}{q(\mathbf{x}_t|\mathbf{x}_{1:t-1})\pi_{t-1}(\mathbf{x}_{1:t-1})})^2] - 1 \tag{26}$$

Minimizing the variance is therefore equivalent to minimizing $\mathbb{E}_{q,\pi_{t-1}}[(\frac{\pi_t(\mathbf{x}_{1:t})}{p(\mathbf{x}_t|\mathbf{x}_{1:t-1})\pi_{t-1}(\mathbf{x}_{1:t-1})})^2]$. Subject to the constraint of the probability, we introduce the Lagrange multiplier $\lambda$ in our objective:

$$\min_{\pi_{t-1}} \mathbb{E}_{q,\pi_{t-1}}[(\frac{\pi_t(\mathbf{x}_{1:t})}{q(\mathbf{x}_t|\mathbf{x}_{1:t-1})\pi_{t-1}(\mathbf{x}_{1:t-1})})^2] + \lambda(\sum_{\mathbf{x}_{1:t-1}} \pi_{t-1}(\mathbf{x}_{1:t-1}) - 1)$$

$$= \min_{\pi_{t-1}} \sum_{\mathbf{x}_{1:t}} \frac{\pi_t(\mathbf{x}_{1:t})^2}{q(\mathbf{x}_t|\mathbf{x}_{1:t-1})\pi_{t-1}(\mathbf{x}_{1:t-1})} + \lambda(\sum_{\mathbf{x}_{1:t-1}} \pi_{t-1}(\mathbf{x}_{1:t-1}) - 1). \tag{27}$$

Taking $\frac{(\cdot)}{\pi_{t-1}(\mathbf{x}_{1:t-1})} = 0$, we get

$$-\sum_{x_t} \frac{\pi_t(\mathbf{x}_{1:t})^2}{q(\mathbf{x}_t|\mathbf{x}_{1:t-1})\pi_{t-1}^2(\mathbf{x}_{1:t-1})} + \lambda = 0. \tag{28}$$

This yields the optimal $\pi_{t-1}^q$ given $q$ as

$$\pi_{t-1}^q(\mathbf{x}_{1:t-1}) = \frac{1}{\lambda}\sqrt{\sum_{\mathbf{x}_t} \frac{\pi_t(\mathbf{x}_{1:t})^2}{q(\mathbf{x}_t|\mathbf{x}_{1:t-1})}}, \tag{29}$$

where $\lambda$ is chosen to normalize the densities to have a sum of 1. Especially, when $\pi_t$ and q correspond to the optimal choices in Equation 13, we recover

$$\pi_{t-1}^q(\mathbf{x}_{1:t-1}) = \frac{1}{\lambda}\sqrt{\sum_{\mathbf{x}_t} \sigma(\mathbf{x}_{1:t})\sigma(\mathbf{x}_{1:t-1})} = \sigma(\mathbf{x}_{1:t-1}) = \pi_{t-1}^*(\mathbf{x}_{1:t-1}). \tag{30}$$

$\square$

### A.3 Proof for the PRM based on Step Correctness

Although the human-annotated process supervision is informally understood as the correctness of a step, there lacks a formal definition of this concept in existing literature (Uesato et al., 2022; Lightman et al., 2024; Wang et al., 2023a; Luo et al., 2024). Here we formally establish the definition of step correctness as follows.

**Definition A.1.** *Each step $\mathbf{x}_t|\mathbf{x}_{1:t-1}$ is either correct or incorrect, following the two axioms below:*

- *For any solution $\mathbf{x}_{1:T}$, if $\mathbf{x}_t|\mathbf{x}_{1:t-1}$ is correct for $t = 1, \cdots, T$, then $\phi(\text{Ans}(\mathbf{x}_{1:T})) = 1$.*

- *For any solution $\mathbf{x}_{1:T}$, if any step $\mathbf{x}_t|\mathbf{x}_{1:t-1}$ is incorrect, then $\phi(\text{Ans}(\mathbf{x}_{1:T})) = 0$.*

Based on Definition A.1 we proceed to prove Proposition 3.3.

**Proposition 3.3.** *The PRM corresponding to the step correctness can be expressed as*

$$r_{PRM}(\mathbf{x}_t|\mathbf{x}_{1:t-1}) = \mathbb{I}(\sigma(\mathbf{x}_{1:t}) > 0). \tag{20}$$

*Proof.* According to Definition A.1, a step is correct as long as it is contained by at least one correct solution. Otherwise, it would be incorrect. Therefore, the process reward in this case corresponds to

$$r_{PRM}(\mathbf{x}_t|\mathbf{x}_{1:t-1}) = \mathbb{I}(\sum_{x_{t+1:T}} \sigma(\mathbf{x}_{1:T}) > 0) = \mathbb{I}(\sigma(\mathbf{x}_{1:t}) > 0). \tag{31}$$

Here, we treat all steps as incorrect if they follow an incorrect step. In Lightman et al. (2024), a step could still be labeled correct even if its previous steps are incorrect. This is because logical thinking is not always in a linear dependency, i.e. a future step is not necessarily dependent on all steps prior to it. However, how to label these steps is up to inductive bias, which does not affect the solution score in the theoretical optimal case.

$\square$

## B Pseudocode for TSMC

Here we summarize the pseudocode for our TSMC-based verification method, where $\text{CONCAT}(\cdot)$ represents the concatenation function, i.e., appending a new element to a list.

---

**Algorithm 1** TSMC for Verification

---

1: **Input**: Generator $p$, estimated value function $V^\theta$
2: **for** $t = 1, \ldots, T$ **do**
3:    **for** $i = 1, \cdots, N$ **do**
4:       # Sample the next step
5:       $\mathbf{x}_t^i \sim p(\cdot | \mathbf{x}_{1:t-1}^i)$
6:       # Concatenate the sampled step to the partial sequence
7:       $\mathbf{x}_{1:t}^i \leftarrow \text{CONCAT}(\mathbf{x}_{1:t-1}^i, \mathbf{x}_t^i)$
8:       # Evaluate the incremental importance weight
9:       **if** $t < T$ **then**
10:          $w(\mathbf{x}_{1:t}^i) \leftarrow \sqrt{\frac{V^\theta(\mathbf{x}_{1:t}^i)}{V^\theta(\mathbf{x}_{1:t-1}^i)}}$
11:       **else**
12:          $w(\mathbf{x}_{1:T}^i) \leftarrow \frac{V^\theta(\mathbf{x}_{1:T}^i)}{\sqrt{V^\theta(\mathbf{x}_{1:T-1}^i)}}$
13:       **end if**
14:    **end for**
15:    **if** $t < T$ **then**
16:       **for** $i = 1, \cdots, N$ **do**
17:          # Resample the sequences
18:          $\omega^i \sim \text{Cat}(\{\frac{w_t(\mathbf{x}_{1:t}^i)}{\sum_{j=1}^N w_t(\mathbf{x}_{1:t}^j)}\}_{i=1}^N)$
19:          $\mathbf{x}_{1:t}^i \leftarrow \mathbf{x}_{1:t}^{\omega_i}$
20:       **end for**
21:    **end if**
22: **end for**
23: # Create a dictionary to store the voting weight
24: $W \leftarrow \{\}$
25: **for** $i = 1, \cdots, N$ **do**
26:    # Extract the answer
27:    $a^i \leftarrow \text{Ans}(\mathbf{x}_{1:T}^i)$
28:    **if** $a^i \in W$ **then**
29:       # Update the answer voting weight
30:       $W[a^i] \leftarrow W[a^i] + w(\mathbf{x}_{1:T}^i)$
31:    **else**
32:       $W[a^i] \leftarrow w(\mathbf{x}_{1:T}^i)$
33:    **end if**
34: **end for**
35: # Voting
36: **return**: $\arg\max_a W[a]$

---

## C   ADDITIONAL EXPERIMENTAL DETAILS

We conducted our experiments for Llemma-7B on 8 NVIDIA H100 GPUs, and our experiments for DeepSeek-7B on 4 NVIDIA RTX A6000 GPUs. Since Llemma-7B is generally weaker than DeepSeek-7B, it in general requires more training samples per problem (the batch size $B$) to obtain at least one positive solution. The summary of the model hyperparameters is presented in Table 2, and we include individual details as follows.

### C.1   GENERATOR TRAINING

We follow Sun et al. (2024) to fine-tune the generators on a filtered subset from PRM800K (Lightman et al., 2024). The hyperparameters are kept the same across the fine-tuning over Llemma-7B (Azerbayev et al., 2023) and DeepSeek-7B (Shao et al., 2024). The generators are fixed once the supervised fine-tuning is over and no additional reinforcement learning is applied.

During the inference time, we generate the solution using top-$K$ sampling with $K = 20$ and set the temperature as 0.7. The maximum length of the solution is fixed as 768.

Table 2: The summary of training hyperparameters for all models.

| **Llemma-7B** | Generator | Value | ORM | PRM (PRM800k) | PRM (SHEPHERD) |
|---|---|---|---|---|---|
| Learning rate | $2 \times 10^{-5}$ | $10^{-5}$ | $2 \times 10^{-5}$ | $2 \times 10^{-5}$ | $2 \times 10^{-5}$ |
| Batch size | 128 | 40 | 128 | 128 | 128 |
| # Epochs | 3 | 2 | 2 | 2 | 2 |
| Warmup ratio | 0.2 | 0.05 | 0.2 | 0.2 | 0.2 |
| Max. length | 768 | | | | |
| Dtype | BF16 | | | | |
| **Deepseek-7B** | Generator | Value | ORM | PRM (PRM800k) | PRM (SHEPHERD) |
| Learning rate | $2 \times 10^{-5}$ | $5 \times 10^{-5}$ | $2 \times 10^{-5}$ | $2 \times 10^{-5}$ | $2 \times 10^{-5}$ |
| Batch size | 128 | 80 | 128 | 128 | 128 |
| # Epochs | 3 | 2 | 2 | 2 | 2 |
| Warmup ratio | 0.2 | 0.10 | 0.2 | 0.2 | 0.2 |
| Max. length | 768 | | | | |
| Dtype | BF16 | | | | |

## C.2 VALUE NETWORK TRAINING

For each math problem in the training dataset, we generate $B$ (the batch size in the above tables) solutions independently with the generator and ignore the problem during training time if all solutions are correct or incorrect. The inference hyperparameters for the generator are kept the same as above. We use the same way to create the validation set using 500 validation instances.

We apply CTL loss (Zhao et al., 2024) on the step-level. The steps are separated by double newline indicators, that is, \n\n, and the value function is trained on the token corresponding to the second newline indicator, along with the end-of-sentence token <eos>. Since the CTL loss is computed over all solutions to a single problem, we fill each training batch with all 80 samples collected from that problem.

Recall that the gradient of the CTL loss at $t$-th step is given by

$$\mathbb{E}_{\sigma(\mathbf{x}_{1:t})}[\nabla_\theta \log V^\theta(\mathbf{x}_{1:t})] - \mathbb{E}_{\pi_t^\theta(\mathbf{x}_{1:t})}[\nabla_\theta \log V^\theta(\mathbf{x}_{1:t})]. \tag{32}$$

We approximate the gradient in the first term via rejection sampling, while the gradient in the second term via IS. The first term is approximated as $\sum_{i=1}^B \frac{\phi(\mathtt{Ans}(\mathbf{x}_{1:T}^i))}{\sum_{j=1}^B \phi(\mathtt{Ans}(\mathbf{x}_{1:T}^j))} \nabla_\theta \log V^\theta(\mathbf{x}_{1:t}^i)$. In the second term, we first compute the importance weight using the current approximated value function $w_t^\theta(\mathbf{x}_{1:t}^i) = \frac{V^\theta(\mathbf{x}_{1:t}^i)}{V^\theta(\mathbf{x}_{1:t-1}^i)}$, then approximate the expected gradient via IS as $\sum_{i=1}^B \frac{w_t^\theta(\mathbf{x}_{1:t}^i)}{\sum_{j=1}^B w_t^\theta(\mathbf{x}_{1:t}^j)} \nabla_\theta \log V^\theta(\mathbf{x}_{1:t}^i)$. Therefore, we can approximate the gradient of $\theta$ on the training problems as

$$\nabla_\theta L_{CTL}(\theta) \approx \mathbb{E}_{\mathbf{x}_0}[\sum_{t=1}^T \sum_{i=1}^B (\frac{\phi(\mathbf{x}_{1:T}^i)}{\sum_{j=1}^B \phi(\mathbf{x}_{1:T}^j)} - \frac{w_t^\theta(\mathbf{x}_{1:t}^i)}{\sum_{j=1}^B w_t^\theta(\mathbf{x}_{1:t}^j)}) \nabla_\theta \log V^\theta(\mathbf{x}_{1:t}^i)]. \tag{33}$$

## C.3 ORM TRAINING

To ensure a fair comparison, the ORM is trained on the same data used to train and validate our value function, but with a different data processing strategy and training method.

We basically follow the same procedure in Cobbe et al. (2021) to train the ORM. We balance the positive and negative samples in the dataset by selecting the same number of correct and incorrect

solutions per problem. The ORM is trained with the binary cross-entropy loss on each token while only the last token is used for prediction during the inference time.

### C.4 PRM TRAINING

We use PRM800K (Lightman et al., 2024) and MATH-SHEPHERD (Wang et al., 2023a) datasets to train two PRMs separately. Especially, we use the PRM800K data to train the PRM once and apply it on both GSM8K (Cobbe et al., 2021) and MATH (Hendrycks et al., 2021) datasets. While for MATH-SHEPHERD, it consists of the samples from both the GSM8K and MATH datasets, and we use the corresponding portion to train two PRMs separately. A validation set of $4096$ samples is held from the training set of each benchmark. We apply the binary cross entropy loss on the second newline token of a step and the last token of each sample.

## D   ADDITIONAL EXPERIMENTS

In this section, we present four additional experiments to check TSMC's advantage over other non-sampling methods, its performance on easy math problems, its generalizability to other reasoning tasks, and its performance on other problems in the MATH dataset (Hendrycks et al., 2021).

### D.1   COMPARISON WITH NON-SAMPLING ALGORITHMS

Although TSMC produces an unbiased estimation of importance weight with less variance, the ultimate goal in reasoning tasks is to generate the solution with the highest probability of correctness. In case we have a perfect estimation of the value function, we could greedily select the partial sequence with the highest value and continue the search since then, which in theory should give the highest chance of reaching the correct answer (Mudgal et al., 2024). But in practice, the training error is unavoidable, and the estimation of the value function is always imperfect, which makes the sampling necessary. In this section, we compare our TSMC method with some non-sampling approaches to verify the necessity of sampling.

The most straightforward non-sampling variant of our TSMC method is blockwise best-of-$K$, (Mudgal et al., 2024), which iteratively selects the partial sequence with the highest value among $K$ partial sequences, and clone the sequence for $K$ times. Such a process is very similar to the resampling process but with the density fully concentrated a single sample. A less greedy method is the Step-Level Beam Search (SBS) (Chen et al., 2024). At each step, SBS selects the top-$B_1$ partial sequences with the highest values, then clones each sample for $B_2$ times to continue the search in the next step. It is also shown to outperform the Monte Carlo Tree Search method (Kocsis & Szepesvári, 2006; Coulom, 2006; Silver et al., 2016; Świechowski et al., 2023) in both the efficiency and solving rate. We follow Chen et al. (2024) to fix $B_2$ in SBS. All these variants share the same batch size for inference as in TSMC, with $K = B_1 B_2 = M = 40$. We fix the value function for all methods and compare TSMC to these variants in Figure 5.

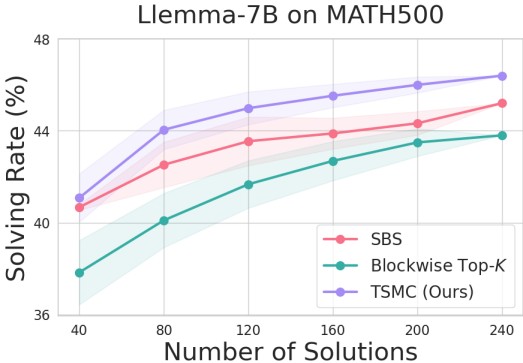

Figure 5: Comparison with non-sampling algorithms. Variance are visualized across many subsamples of the 240 solutions per problem.

It can be seen that our TSMC method still consistently outperforms both non-sampling approaches. Although SBS achieves better results than other baselines in Table 1, there remains a clear performance gap compared to our TSMC method. This underscores the robustness of the TSMC method to the estimation error of the value function and highlights the importance of sampling in handling such an error.

## D.2 EASY MATH PROBLEMS

To examine whether TSMC has an adverse effect when the problem is easy, we evaluate the performance of TSMC on an easy math reasoning benchmark, Multiarith (Roy & Roth, 2015). Since the answer format of Multiarith is identical to that of GSM8K (Cobbe et al., 2021), consisting of integer numbers, we directly utilize the generators and reward/value models trained on GSM8K to evaluate this dataset. All experimental setups remain consistent with those described in Section 4. The comparative results are shown in Table 3.

Table 3: Comparative results in the problem solving rate (%) on the Multiarith dataset. We use two generators fine-tuned from pretrained Llemma-7B and DeepSeek-7B, respectively. We bold the best results in each category. The voting is performed on 240 samples.

| Generators | Methods | Multiarith |
|---|---|---|
| Llemma-7B | Greedy | 66.7 |
| | MV | 95.4 |
| | WMV w. ORM | 97.1 |
| | WMV w. PRM (PRM800K) | 95.4 |
| | WMV w. PRM (SHEPHERD) | 97.8 |
| | TSMC + MV (Ours) | 98.3 |
| | TSMC + WMV (Ours) | **98.9** |
| DeepSeek-7B | Greedy | 85.6 |
| | MV | 98.3 |
| | WMV w. ORM | 98.3 |
| | WMV w. PRM (PRM800K) | 97.7 |
| | WMV w. PRM (SHEPHERD) | **98.9** |
| | TSMC + MV (Ours) | **98.9** |
| | TSMC + WMV (Ours) | **98.9** |

Although the absolute performance gap between TSMC and other baselines is reduced, TSMC, especially TSMC + MV, still shows a consistent improvement without showing any adverse effect. But in practice, the usage of TSMC should balance the trade-off between the performance improvement and the additional computational cost. In general, TSMC is particularly well-suited for scenarios where the generator lacks the capacity to produce correct solutions.

## D.3 OTHER REASONING TASK

In this section, we demonstrate the generalizability of TSMC to other reasoning tasks beyond mathematical problems. Here we choose the quantitative natural language inference task in the NumGLUE benchmark (Mishra et al., 2022), which uses a Python program for multi-step reasoning.

We separate the Python program into steps using the single newline character \n. We re-split the dataset by choosing a subset of the original validation and testing sets for validation and testing, and add the remaining ones to the training set. This forms a final dataset with 5924 training samples, 200 validation samples, and 200 testing samples. The training set is used for the fine-tuning of the generator and the estimation of the value function, with CodeLlama-7B (Rozière et al., 2024) as the pretrained model. The entire training process is kept the same as before. Since this task is relatively easy, we perform TSMC across the full reasoning process, without skipping the first few tokens or setting the maximum number of resampling steps.

The comparative results are presented in Table 4 below. Since PRMs are not available for this task, we only compare our method with zero-shot greedy decoding, majority voting, and the ORM

method. Here we vary the voting sample size across $\{5, 10, 20, 40\}$, and the TSMC batch size is set the same as the number of voting samples. The results indicate that TSMC achieves a consistent advantage over the baselines in all voting sizes. It still achieves a high solving rate (99%) with only 5 voting samples. Despite the simplicity of this task, the promising results have clearly demonstrated the potential of TSMC for application to other reasoning tasks.

Table 4: Comparative results in the problem solving rate (%) on the NumGLUE dataset (quantitative natural language inference). We fine-tune our generator from pretrained CodeLlama-7B. The voting is performed under $N = 5, 10, 20$ and $40$ respectively (no difference for the zero-shot greedy decoding). We bold the best results in each category.

| Generator | Methods | $N = 5$ | $N = 10$ | $N = 20$ | $N = 40$ |
|---|---|---|---|---|---|
| CodeLlama-7B | Greedy | 94.0 | 94.0 | 94.0 | 94.0 |
| | MV | 97.0 | 98.0 | 98.0 | 98.0 |
| | WMV w. ORM | 97.0 | 98.5 | 98.5 | 98.5 |
| | TSMC + MV (Ours) | 98.5 | 99.0 | 99.5 | 99.5 |
| | TSMC + WMV (Ours) | **99.0** | **99.5** | **99.5** | **99.5** |

### D.4 OTHER PROBLEMS IN THE MATH DATASET

To investigate whether the conclusion on MATH500 could be generalized to other problems in the MATH dataset (Hendrycks et al., 2021), here we repeat our evaluation on the on-hold validation set, which consists of $500$ samples as well. The results are shown in Table 5. It can be seen that the results are basically consistent with the ones presented in Table 1, where TSMC takes a consistent lead over all baselines.

Table 5: Comparative results in the problem solving rate (%) on the on-hold validation set in MATH. We use two generators fine-tuned from pretrained Llemma-7B and DeepSeek-7B, respectively. We bold the best results in each category. The voting is performed on 240 samples.

| Generators | Methods | MATH (validation) |
|---|---|---|
| Llemma-7B | Greedy | 26.2 |
| | MV | 40.2 |
| | WMV w. ORM | 42.4 |
| | WMV w. PRM (PRM800K) | 44.0 |
| | WMV w. PRM (SHEPHERD) | 46.8 |
| | TSMC + MV (Ours) | 47.4 |
| | TSMC + WMV (Ours) | **50.8** |
| DeepSeek-7B | Greedy | 33.2 |
| | MV | 55.2 |
| | WMV w. ORM | 57.2 |
| | WMV w. PRM (PRM800K) | 58.4 |
| | WMV w. PRM (SHEPHERD) | 56.4 |
| | TSMC + MV (Ours) | 56.8 |
| | TSMC + WMV (Ours) | **61.6** |

## E QUALITATIVE EXAMPLES

Here we include the qualitative examples generated by our TSMC method in Table 6. We compare the samples discarded (do not get sampled) and cloned by TSMC (get sampled multiple times) during the same resampling step. We also include the estimated value and the incremental importance weight in each sample. Notably, the incremental importance weight is proportional to the resampling probability up to a constant, so a lower weight means a lower chance to be sampled.

| | |
|---|---|
| QUESTION | What is $1^{\left(2^{235423523}\right)}$? |
| DISCARDED

VALUE: 0.09
WEIGHT: 0.77 | I notice that this expression is a power of 1, so it is just 1 by the definition of exponents.

However, I also notice that the exponent is very large, so I wonder if there is a pattern or a shortcut to simplify it.

I recall that any number raised to a multiple of 2 has a simple pattern: it is either 1 or $-1$ depending on whether the exponent is even or odd. |
| CLONED

VALUE: 0.18
WEIGHT: 1.25 | So the question is asking us to evaluate $1^{\left(2^{235423523}\right)}$.

Right, and we know that anything to the power of 1 is just 1.

So now we just need to evaluate $1^{2^{235423523}}$.

And since 1 is a power of 1, then $1^{2^{235423523}} = 1$. |
| QUESTION | A regular hexagon can be divided into six equilateral triangles. If the perimeter of one of the triangles is 21 inches, what is the perimeter, in inches, of the regular hexagon? |
| DISCARDED

VALUE: 0.02
WEIGHT: 0.27 | Since the hexagon is regular, all the sides are equal.

And since the hexagon can be divided into six equilateral triangles, all the sides of the hexagon are also equal to the perimeter of one of the triangles divided by six.

So the perimeter of the hexagon is $21/6 = 3.5$ inches. |
| CLONED

VALUE: 0.36
WEIGHT: 1.37 | I know that the perimeter of a regular hexagon is six times the length of one of its sides, so I need to find the length of one side of the hexagon.

If I divide a regular hexagon into six equilateral triangles, each triangle must have the same side length as the hexagon, because they are congruent.

So, if the perimeter of one of the triangles is 21 inches, then each side of the triangle must be $21/3 = 7$ inches. |
| QUESTION | How many positive whole-number divisors does 196 have? |
| DISCARDED

VALUE: 0.17
WEIGHT: 0.85 | To find the number of divisors of a positive whole-number, I can use the fact that it is equal to the product of one more than the exponents of its prime factorization.

For example, 12 has six divisors because it can be written as $2^2 * 3^1$, and the product of one more than the exponents is $(2 + 1) * (1 + 1) = 6$.

So I need to find the prime factorization of 196

I can start by dividing by 2, the smallest prime, and see if I get an integer.

$196/2 = 98$, which is not an integer, so 2 is not a factor of 196.

I can try the next prime, 3, and see if I get an integer. |
| CLONED

VALUE: 0.33
WEIGHT: 1.10 | I need to find all the positive integers that divide 196 evenly, without leaving a remainder.

One way to do this is to factor 196 into its prime factors, and then use the formula for the number of divisors of a product of prime powers.

196 is divisible by 2, so I can write it as 2 times 98.

98 is also divisible by 2, so I can write it as 2 times 49.

49 is a perfect square, and it is also divisible by 7, so I can write it as 7 times 7.

Therefore, 196 can be written as 2 times 2 times 7 times 7, or $2^2$ times $7^2$. |

Table 6: Qualitative examples for partial sequences discarded and cloned by TSMC. The estimated value and incremental importance weight from the TSMC are included. We also manually annotate the incorrect/unpromising reasoning steps and highlight them in the pink color.

# F    EXAMPLE OUTPUTS

Here we include some example outputs from the generator, which follows the basic format in
PRM800K (Lightman et al., 2024). `# Question` and `# Solution` indicate the input problem
and the generated solution, respectively.

```
# Question

Convert the point $(0,3)$ in rectangular coordinates to polar
    coordinates. Enter your answer in the form $(r,\theta),$ where $r > 0
    $ and $0 \le \theta < 2 \pi.$

# Solution

To convert from rectangular to polar coordinates, I need to use the
    formulas $r = \sqrt{x^2 + y^2}$ and $\theta = \tan^{-1} \frac{y}{x}.$

For this point, $x = 0$ and $y = 3,$ so $r = \sqrt{0^2 + 3^2} = 3$ and $
    \theta = \tan^{-1} \frac{3}{0}.$

However, I can't divide by zero, so I need to use a limit to avoid that.

As the angle $\theta$ approaches zero from the right, the ratio $\frac{y
    }{x}$ approaches $\frac{3}{0^+}.$

This means that $\theta$ approaches $\frac{\pi}{2},$ which is the angle
    in the first quadrant.

Since $r$ is always positive, I don't need to worry about the sign or
    the quadrant.

Therefore, the polar coordinates are $(3, \frac{\pi}{2}).$

# Answer

(3, \frac{\pi}{2})
```

```
# Question

The expression $2\cdot 3 \cdot 4\cdot 5+1$ is equal to 121, since
    multiplication is carried out before addition. However, we can obtain
     values other than 121 for this expression if we are allowed to
    change it by inserting parentheses. For example, we can obtain 144 by
     writing \[
(2\cdot (3\cdot 4)) \cdot (5+1) = 144.
\]In total, how many values can be obtained from the expression $2\cdot
    3\cdot 4 \cdot 5 + 1$ by inserting parentheses? (Note that
    rearranging terms is not allowed, only inserting parentheses).

# Solution

I need to count how many ways I can group the four factors of the
    product into two pairs, and then combine those pairs using either
    multiplication or addition.

I can use the binomial coefficient to count the number of ways to choose
     two factors out of four, and then multiply that by the number of
    ways to group the remaining two factors.

The binomial coefficient $\binom{4}{2} = \frac{4!}{2!2!}

= 6$ is the number of ways to choose two factors out of four, and it
    represents the number of ways to arrange the two pairs of factors.
```

```
The number of ways to combine those pairs into a single product or sum
    depends on which pair I choose to be the first one.

If I choose the pair $2\cdot 3$, then I have two options: either to
    multiply them $(2\cdot 3)$, or to add them to the rest of the
    expression $(2\cdot 3) \cdot 4\cdot 5 + 1$.

If I choose the pair $4\cdot 5$, then I have two options: either to
    multiply them $(4\cdot 5)$, or to add them to the rest of the
    expression $(2\cdot 3) \cdot 4 + 1$.

In either case, the rest of the expression $(4\cdot 5) \cdot 2\cdot 3 +
    1$ or $(2\cdot 3) \cdot 4 + 1$ is always a constant, and I can ignore
     it for now.

So, for each pair of factors, I have two options to combine them into a
    product or a sum.

Therefore, for each way to arrange the two pairs of factors, I have $2\
    cdot 2 = 4$ options to combine them into a product or a sum.

This means that there are $6\cdot 4 = 24$ ways to insert parentheses in
    the expression, and obtain values other than 121.

# Answer

24
```