# OpenReview forum: "Step-by-Step Reasoning for Math Problems  via Twisted Sequential Monte Carlo"
_ICLR.cc/2025/Conference — ICLR 2025 Poster_

### Official Review · Reviewer_bjdp · 2024-11-03

**Soundness:** 3
**Presentation:** 3
**Contribution:** 3
**Rating:** 8
**Confidence:** 3

**Summary:**

This paper explores methods to improve sample efficiency in verification to enhance step-by-step math reasoning using Large Language Models (LLMs). The main innovation is a "twisted" variant of Sequential Monte Carlo (TMSC), which applies re-weighting in importance sampling at each step of solution generation. Experimental results show that this approach demonstrates promise on two math datasets, GSM8K and MATH500 (a subset of 500 representative examples from the MATH dataset), compared to baseline methods such as majority voting and verification-based approaches. The verification-based baselines fall into two categories: those that evaluate performance on the full solution (outcome reward model) and those that evaluate performance at each intermediate step (process reward model).

**Strengths:**

1. Addressing the challenge of improving LLM-based math reasoning is timely.
2. The paper is well-written and effectively positions its contributions within the current state-of-the-art.
3. The use of twisted SMC for this problem is principled and novel, with experimental results highlighting the strengths of this approach in improving the problem-solving rate over the baseline.
4. TSMC effectively combines the advantages of unbiased estimation from the ORM and intermediate step modeling from the PRM.

**Weaknesses:**

1. The MATH500 dataset is a selection of 500 problems from the larger MATH dataset of 12,500 problems. It is unclear how well the approach will scale or perform on the full dataset.

**Questions:**

1. How were the 500 problems selected from the 12.5K problems in the MATH dataset? Is performance expected to remain consistent across the complete dataset?

---

> ### Author Response · Authors · 2024-11-23
> **Response**
>
> We sincerely appreciate your recognition of the significance and novelty of our work. Here we mainly address your concern in terms of the MATH500 dataset.
>
> > How were the 500 problems selected from the 12.5K problems in the MATH dataset? Is performance expected to remain consistent across the complete dataset?
>
> Following [1], we utilize the 500 problems (the same split in [1]) uniformly sampled from the whole dataset as the test set. This is a common practice employed in existing works [1][2][3].
>
> Since we have already trained our model on most remaining problems, we verify the performance of our method on our on-hold validation dataset (500 samples). The result is presented in Appendix G Table 6, which is consistent with our result on the testing set.
>
> [1] Lightman et al. Let’s verify step by step. ICLR 2024.
>
> [2] Wang et al. Math-Shepherd: Verify and Reinforce LLMs Step-by-step without Human Annotations. ACL 2024.
>
> [3] Sun et al. Easy-to-hard generalization: Scalable alignment beyond human supervision. NeurIPS 2024.

---

### Official Review · Reviewer_16x7 · 2024-11-04

**Soundness:** 3
**Presentation:** 4
**Contribution:** 3
**Rating:** 6
**Confidence:** 4

**Summary:**

This paper presents Twisted Sequential Monte Carlo (TSMC), designed to enhance the efficiency of verification in LLMs for multi-step reasoning tasks. Unlike current verification approaches, TSMC refines its sampling effort progressively, focusing on promising solution candidates. It estimates expected future rewards for partial solutions, enabling a more efficient generation of high-quality outputs without step-wise human annotations.

**Strengths:**

This paper introduces Twisted Sequential Monte Carlo to enhance the verification of multi-step reasoning in LLMs. It redefines verification tasks as an importance sampling problem, providing a novel perspective. The methodology is thoughtfully developed with theoretical analysis, addressing sampling inefficiencies in existing approaches. The writing is clear and well-organized.

**Weaknesses:**

The method demonstrates promise but still faces some challenges. The reliance on a simplified proposal distribution and greedy optimization may introduce sampling bias, increasing the risk of converging to local optima in complex tasks. Additionally, some key implementation details are lacking, such as strategies for handling resampling limits. The absence of inference-time statistics raises concerns about potential computational overhead.
Moreover, it heavily depends on sampling efficiency improvements without a full exploration of how TSMC affects other crucial aspects of performance, such as generalizability and robustness to different problem types. In other words, it’s focus on reducing sampling inefficiency and process supervision requirements addresses only specific limitations in existing verification methods, which may limit the broader applicability of TSMC across other reasoning tasks. The paper's evaluation is also primarily limited to mathematical benchmarks (such as GSM8K and MATH500), making it challenging to generalize findings to diverse reasoning contexts, such as logical inference or more complex free form problems. Further,, more comprehensive empirical validation in varied real-world scenarios could benefit the paper.
Finally, code or data are not provided.

**Questions:**

1) The propose methods in Section 3.2 partially address the challenges in TSMC, but there are still issues which could impact TSMC’s sampling quality and efficiency, especially in complex and high-dimensional tasks. For example, 1. the proposal distribution $q(x_t | x_{1:t-1})$ is simplified to $p(x_t | x_{1:t-1})$. It may cause samples to deviate from the target distribution, introducing sampling bias. 2. Using a greedy strategy to iteratively optimize the intermediate target distribution can lead to a local optimum problem. 3. Even with the recursive optimization strategy, excessive variance remains a challenging issue in high-dimensional spaces. Do the authors consider further improvements, such as  global optimization strategies?


2) The solution in Section 3.4 relies on the accurate estimation of the value function $V^{\theta}$. However, learning the value function can often be unstable or divergent in many cases. In particular, the error accumulation problem is more likely to happen in multi-step reasoning paths, which may cause the model to deviate during the reasoning process. Do the authors make any improvements or additional designs here?


3) Some key implementation details are missing. Specifically, 1. The maximum number of resampling steps is set to 5. However, the selection strategy when the number of steps exceeds this limit is not discussed. 2. Baseline performance of the two generators (without majority voting, using only chain-of-thought) should be provided to better contextualize the improvements. 3. The reward model employed in the TSMC + WMC strategy is not specified.


4) While TSMC enhances sampling efficiency through resampling, it may also introduce substantial computational overhead during inference. Could the authors provide inference time statistics for the proposed method? Please include discussion on  the computational overheads associated with implementing TSMC, especially concerning memory usage, processing time, and scalability? Further, would these requirements limit TSMC’s practical use for large-scale or real-time applications?

5) please include explanation as to how the method can extend beyond the narrow focus on mathematical problem-solving, such as those in logical reasoning and free form reasoning problems that do not have structure.

6) Can the authors provide a deeper analysis of TSMC’s failure cases or error patterns?

7) The code and data are not provided for reproducibility. The paper can benefit significantly from providing those.

---

> ### Author Response · Authors · 2024-11-23
> **Response (1/3)**
>
> Thank you for your detailed and insightful questions on our work. Basically, we understand the concerns in some of the designs in our method, but we would like to remind the reviewer that all our designs have been verified through our empirical results. Here we provide a more intuitive explanation in why these designs work.
>
> > the proposal distribution  $q(x_t|x_{1:t−1})$ is simplified to $p(x_t|x_{1:t−1})$. It may cause samples to deviate from the target distribution, introducing sampling bias
>
>
>
> Our method is primarily an  inference-time search algorithm and that is why we assume we only have access to a poor generator $p(x_t|x_{1:t-1})$. Similar assumption is also used in other verification methods such as [1].
>
> However, this does not mean that we shall always stick to $q=p$. A better generator (e.g., via finetuning) would definitely improve the final performance of any verification method.
>
>
> [1] Lightman et al. Let’s verify step by step. ICLR 2024.
>
> > Using a greedy strategy to iteratively optimize the intermediate target distribution can lead to a local optimum problem
> > Do the authors consider further improvements, such as global optimization strategies?
>
> We would like to kindly point out that TSMC is a sampling process rather than an optimization process, so the local optimum in the variance reduction would not lead to the “local optimum” in the generated samples.
>
> In standard verification methods, there is no optimization of the variance, and in TSMC, we reduce the variance via informative intermediate targets. Although they are not globally optimal, they already take effect compared with standard verification methods.
>
> When applying the Lagrange multiplier (Section A.2) to find the global optimum by zeroing the derivatives, it ends up with solving an ordinary differential equation with two boundary conditions at $t=1$ and $t=T$ and turns out to be unsolvable. Therefore, the globally optimal intermediate targets are not in a clean formulation and it is hard to approximate them in practice. But even with the global optimum, the variance would not be reduced to 0 unless the generative distribution is the same as the marginal target. So our locally optimal targets have already been a practically good choice under our problem setting.
>
> > However, learning the value function can often be unstable or divergent in many cases.
> > Do the authors make any improvements or additional designs here?
>
>
> We acknowledge that the value estimation could be unstable, but we would also like to point out this is exactly the advantage of TSMC against the greedy decoding method (concentrating all explorations on the partial sequence with the highest value).
>
> In TSMC, if the value function is uninformative (not trained), say a constant, the stratified sampling strategy we used would simply make TSMC the same as the standard verification method without resampling. We also skip the first few tokens since the value estimation there could be inaccurate. We summarize these strategies in Section 4.1 TSMC details. In a nutshell, TSMC takes effect if the value estimation is informative and and brings no adverse effect if it is not.

---

> > ### Author Response · Authors · 2024-11-23
> > **Response (2/3)**
> >
> > > Please include discussion on the computational overheads associated with implementing TSMC
> >
> > We want  to thank you for this insightful question. We would like to first point out that the main focus of this paper is on the algorithm development, while the practical use of TSMC involves numerous  infra optimizations to reduce the memory and inference costs, e.g., quantization, kv-cache, dynamic-batching, which are out of the scope of this work. But we are happy to discuss the current limitations and the possible solutions.
> >
> > Since we need to load two models, the time consumption and the memory consumption are around 2 times of the standard parallel decoding methods. We use a batch size 80 and the average time/memory usage for each batch is shown below. For verification baselines, there would be an additional 2 seconds atop the cost from parallel decoding.
> >
> > Time usage:
> >
> > |		             | MATH500 (Llemma)  |MATH500 (DeepSeek) | GSM8K (Llemma)  |GSM8K (DeepSeek) |
> > --- | --- | --- | --- |--- |
> > |Parallel Decoding |  8.53                   |10.99                             | 8.01                        |  9.67 |
> > |TSMC		      | 23.29     	           |  22.34	                      |17.36                       | 15.24 |
> >
> >
> > (The memory usage is hard to measure, we take a snapshot of the memory usage in the middle of the inference)
> >
> > Memory Usage:
> >  |                              | Llemma | DeepSeek|
> > --- | --- | --- |
> > |Parallel Decoding |  36G        | 52G|
> > |TSMC:                  |  70G       |  78G|
> >
> > However, this additional cost from the value network could be easily resolved by using a shared backbone for the generator and the value network, as in the classical actor-critic models.
> >
> > Another source of computational latency comes from the asynchronous decoding of the steps across sequences, where the sequence with a shorter step needs to stop and wait for the sequence with a longer step. One potential solution is to define the step as a fixed number of tokens for a better parallelism, whose performance (denoted as TSMC-block) on MATH500 (Llemma) is shown below.
> >
> > |		             |Time (s) |Solving Rate (%)|
> > --- | --- | --- |
> > |Parallel Decoding|   8.53   | 41.2    |
> > |TSMC         |  23.29  | 45.6  |
> > |TSMC-block         |  15.86  | 45.4|
> >
> >
> > But overall, TSMC does not involve additional generation since the resampling step simply copies the promising sequences to replace the low-quality ones, without generating more candidate steps such as the backtracking. This ensures the potential of TSMC to be further accelerated in real-time applications.
> >
> > > Some key implementation details are missing
> >
> > Thanks for your reminder, here we clarify some details according to your examples.
> > - After the maximum number of resampling steps, there is simply no resampling and the decoding is the same as the standard parallel decoding.
> > - We have added the greedy decoding method in Table 1, thank you for your suggestion.
> > - The reward function used by TSMC is the same as the value function (estimated at the last token). This is mentioned in Section 3.4 and Appendix C.2

---

> > > ### Author Response · Authors · 2024-11-23
> > > **Response (3/3)**
> > >
> > > > please include explanation as to how the method can extend beyond the narrow focus on mathematical problem-solving
> > >
> > > TSMC provides a unified framework for all multi-step reasoning tasks, as long as the “step” could be well-defined.
> > >
> > > Following your suggestion, we included an additional experiment on the NumGLUE dataset for the natural language inference task. The reasoning process is a Python program. Please see Table 5 in Appendix G for the result. Our TSMC method still takes a consistent advantage under various numbers of voting samples. On this dataset, we separate the Python program into steps  via the newline character “\n”
> > >
> > > Beyond steps defined according to delimiters, TSMC can also be applied to steps defined by a fixed number of tokens (as shown above in TSMC-block), enabling it to generalize effectively to any reasoning task. But overall, the reasoning step split is a general research problem, where different splitting strategies are expected from different domains.
> > >
> > > > Can the authors provide a deeper analysis of TSMC’s failure cases or error patterns?
> > >
> > > The first failure case is when the value function is not informative, TSMC takes a similar performance as the standard verification method but with a higher computational cost.
> > >
> > > The second failure case is the risk of the weight degeneracy issue. That is, when the probability of a sample is dominant in the categorical distribution for resampling, most of the newly resampled sequences will come from the same sample, leading to an additional variance.

---

### Official Review · Reviewer_jY8a · 2024-11-04

**Soundness:** 2
**Presentation:** 3
**Contribution:** 3
**Rating:** 6
**Confidence:** 3

**Summary:**

This paper presents a method based on Twisted Sequential Monte Carlo (TSMC) to enhance the multi-step reasoning capabilities of large language models (LLMs) in mathematical problem-solving. TSMC improves the efficiency of generating high-quality solutions by progressively optimizing the sampling process and focusing exploration on promising candidate solutions. The paper also demonstrates the advantages of this method across two mathematical benchmarks and validates its effectiveness through theoretical analysis.

**Strengths:**

The article proposes a new method based on Twisted Sequential Monte Carlo to enhance the sampling efficiency of large models while reducing reliance on process supervision. The work is original, with a solid theoretical foundation, achieving good results on two benchmarks. Additionally, it is easy to follow.

**Weaknesses:**

The article provides ample theoretical analysis; however, I have some questions regarding the effectiveness of the proposed method.
A robust and scalable method should perform well across all datasets, so I am curious whether it can achieve similar results on relatively simpler mathematical datasets(without causing adverse effects), such as Addsub, MultiArith, and SVAMP.

Additionally, I suggest adding a comparison for w/o MV （zero-shot performance）in Table 1, which would provide a better comparison.

**Questions:**

In Figure 4, why does the solving rate via TSMC show continuous and significant improvement with an increasing number of solutions for the same batch size (e.g., M=10)? Is this phenomenon unique to TSMC, or do other comparison algorithms exhibit similar behavior? Please provide more explanation on this point.

In the algorithm pseudocode, is there any difference between 'concat' and 'cat' (Appendix B)?

Could you provide a more detailed comparison with Zhao's work (line 524)?

---

> ### Author Response · Authors · 2024-11-23
> **Response**
>
> Thank you for your recognition of the novelty and theoretical foundation of our work. Here we have included a detailed response with additional experimental results to address your concerns.
>
> > whether it can achieve similar results on relatively simpler mathematical datasets(without causing adverse effects), such as Addsub, MultiArith, and SVAMP.
>
> Thank you for your insightful question. Following your suggestion, we have added a comparative experiment on the MultiArith dataset. Please see Table 4 in Appendix G for the details.
>
> The results show that TSMC outperforms the baselines under all scenarios without bringing any adverse effects. All the inference-time parameters are kept the same over the three datasets (MATH500, GSM8K, Multiarith).
>
> > adding a comparison for w/o MV （zero-shot performance）in Table 1
>
> Thank you for your suggestion, we have added the comparison in Table 1. The method is denoted as “Greedy” , using the greedy decoding strategy.
>
> > In Figure 4, why does the solving rate via TSMC show continuous and significant improvement with an increasing number of solutions for the same batch size (e.g., M=10)
>
> We would like to first clarify that the batch size here refers to the resampling batch size of the TSMC rather than the number of voting samples $N$, where the latter is kept as a constant as 240 in this paper. $M$ is a TSMC-specific parameter not present in other methods.
>
> The resampling batch size of TSMC refers to the number of samples used to form the categorical distribution for resampling in Equation 5. For example, when M=10, we would have 24 distributions and each distribution contains 10 categories.
>
> The optimal number we find in Figure 4 is actually $M=20$. We hypothesize this is related to the weight degeneracy issue of importance sampling. When having a large resampling batch size, one sample with a large probability would dominate the distribution and lead to a bad diversity after resampling. So there is a tradeoff between the diversity and the variance. We treat the understanding of this phenomenon as a part of our future work as stated in Section 6.
>
>
> > In the algorithm pseudocode, is there any difference between 'concat' and 'cat' (Appendix B)?
>
> Yes. ‘Cat’ is initially defined in Equation 5, whose meaning is the categorical distribution. ‘Concat’ simply means the concatenation. We have added one more comment in the pseudocode to clarify this.
>
> > Could you provide a more detailed comparison with Zhao's work (line 524)
>
> Both works follow the classical framework of TSMC, while Zhao’s work is more focused on a general alignment framework while our method is primarily an inference-time search algorithm.
>
>
> The major difference lies in our assumption that  the generator is not powerful enough and refining the distribution of the generator is expensive (intractable step-level guidance), this is in line with the motivation of recently popular  inference-time searching algorithms like MCTS, backtracking or o1 model. With this constraint, we can only use a suboptimal generator $q=p$ and need to address the potential large variance via more informative intermediate targets (Section 3.2).
>
>
> In Zhao’s work, they allow the refinement of the generator via a tractable token-level guidance, but this token-level guidance would be expensive in case of long outputs since it needs the estimation of the twist objective at each token. This makes our work more significant under this setting (long output), for example, the multi-step reasoning task.

---

### Official Review · Reviewer_UXiV · 2024-11-04

**Soundness:** 3
**Presentation:** 1
**Contribution:** 2
**Rating:** 5
**Confidence:** 2

**Summary:**

The paper presents a method to enhance the multi-step reasoning capabilities of Large Language Models (LLMs) by using Twisted Sequential Monte Carlo (TSMC) for verification. Existing verification methods for math problems, such as the Outcome Reward Model (ORM) and Process Reward Model (PRM), face limitations such as low sampling efficiency and high reliance on human-supervised annotations. TSMC refines sampling at each intermediate step, unlike standard methods that evaluate only fully generated solutions. TSMC is tested on two math benchmarks (GSM8K and MATH500) and results indicate that TSMC consistently outperformed existing methods.

**Strengths:**

1. The challenge presented is up-to-date.
2. Empirical results showing that TSMC outperforms existing methods on math benchmarks.

**Weaknesses:**

1. This article mentions two problems in the existing methods, and problem I has been thoroughly discussed in the paper. However, problem II seems to be neglected.
2. The dataset MATH500 is created through selecting 500 problems from the larger MATH dataset. Whether the questions are representative remains to be seen.

**Questions:**

1. How does your method address the problem II?
2. Could you give a statement about how you select your problems from MATH and prove (or show) that your method works well on the whole dataset?

---

> ### Author Response · Authors · 2024-11-23
> **Response**
>
> Thank you for your recognition in our effort to deal with the low-sampling efficiency problem (Problem I) in existing verification methods. Here we would like to clarify our approach in resolving the need for process supervision (Problem II) and the concern of using MATH500.
>
> > How does your method address problem II?
>
> We have declared our solution to Problem II in Section 3.4, “Estimating the value function through independently sampled data from policy (generator) is a well-studied topic (Bertsekas, 2012). It therefore eliminates the need for explicit process supervision during training, as outlined in Problem II.”
>
> Our TSMC method is based on the guidance from the value network. While the training of the value network only requires the data sampled from the generator and the correctness of the solution, without need of the process supervision. Notably, we only need to sample the data from the generator independently using the standard parallel decoding, meaning there is no roll-out from an intermediate step of a sequence as in tree search. So our method exhibits a high training efficiency as well (Problem II).
>
> > How math500 is being selected. How it works on the whole dataset.
>
> Following [1], we utilize the 500 problems (the same split in [1]) uniformly sampled from the whole dataset as the test set. This is a common practice employed in existing works [1][2][3].
>
> Since we have already trained our model on most remaining problems, we verify the performance of our method on our on-hold validation dataset (500 samples). The result is presented in Appendix G Table 6, which is consistent with the result on our testing set.
>
> [1] Lightman et al. Let’s verify step by step. ICLR 2024.
>
> [2] Wang et al. Math-Shepherd: Verify and Reinforce LLMs Step-by-step without Human Annotations. ACL 2024.
>
> [3] Sun et al. Easy-to-hard generalization: Scalable alignment beyond human supervision. NeurIPS 2024.

---

> > ### Author Response · Authors · 2024-12-02
> >
> > Dear reviewer UXiV, as the discussion period is coming to an end, we would like to kindly confirm with you if our response has addressed your concern. Thank you again for your valuable time and hard work in reviewing our work.

---

> > > ### Comment · Reviewer_bjdp · 2024-12-02
> > >
> > > Thank you for reaching out. Your response has adequately addressed my concerns.

---

### Official Review · Reviewer_DqPL · 2024-11-04

**Soundness:** 3
**Presentation:** 3
**Contribution:** 3
**Rating:** 8
**Confidence:** 2

**Summary:**

This paper addresses the challenge of improving multi-step reasoning capabilities in large language models (LLMs) for mathematical problem-solving. Current verification methods, which are employed to assess the accuracy of solutions generated by LLMs, suffer from inefficiencies in sampling and require costly process supervision. The authors propose a novel verification approach using Twisted Sequential Monte Carlo (TSMC) to enhance sampling efficiency by focusing on partial solutions during the generation process. The method reduces reliance on human annotations, making it more scalable. The paper demonstrates the superiority of TSMC through theoretical analysis and empirical results across two math benchmarks.

**Strengths:**

- The use of TSMC in verification for LLM reasoning is novel and shows promise in addressing inefficiencies in existing methods.
- Experiments on GSM8K and MATH500 benchmarks demonstrate that TSMC improves problem-solving rates.
- The approach reduces the need for detailed human process supervision, making it more practical for large-scale applications.

**Weaknesses:**

- Limited discussion on practical implementation challenges or resource requirements for integrating TSMC in real-world systems.
- The paper focuses on mathematical problem-solving; it would be useful to discuss potential applications or limitations in other multi-step reasoning tasks.

**Questions:**

- How does TSMC perform when applied to multi-step reasoning tasks beyond mathematical problems, such as programming or logical puzzles?
- What are the computational costs associated with TSMC in terms of training and inference time compared to existing methods?
- Can the method be adapted for dynamic or variable-length intermediate steps in more complex reasoning tasks?
- How sensitive is the proposed approach to the choice of generator model and dataset size?

---

> ### Author Response · Authors · 2024-11-23
> **Response (1/2)**
>
> We would like to thank you for your positive feedback and constructive suggestions on our work. Please see below for the detailed responses to each of your concerns.
>
> > What are the computational costs associated with TSMC in terms of training and inference time compared to existing methods?
>
> Thank you for your insightful question. Since TSMC needs to load two models simultaneously, its inference computational cost is around 2 times of the existing methods. We use a batch size 80 and the average time usage (in seconds) for each batch is shown below. For verification methods, there would be an additional 2 seconds atop the cost from parallel decoding.
>
> |		             | MATH500 (Llemma)  |MATH500 (DeepSeek) | GSM8K (Llemma)  |GSM8K (DeepSeek) |
> --- | --- | --- | --- |--- |
> |Parallel Decoding |  8.53                   |10.99                             | 8.01                        |  9.67 |
> |TSMC		      | 23.29     	           |  22.34	                      |17.36                       | 15.24 |
>
> However, this additional cost from the value network could be easily resolved by using a shared backbone for the generator and the value network, as in the classical actor-critic models.
>
> Another source of computational latency comes from the asynchronous decoding of the steps across sequences, where the sequence with a shorter step needs to stop and wait for the sequence with a longer step. One potential solution is to define the step as a fixed number of tokens for a better parallelism, whose performance (denoted as TSMC-block) on MATH500 (Llemma) is shown below.
>
> |		             |Time (s) |Solving Rate (%)|
> --- | --- | --- |
> |Parallel Decoding|   8.53   | 41.2    |
> |TSMC         |  23.29  | 45.6  |
> |TSMC-block         |  15.86  | 45.4|
>
> Overall, TSMC does not involve additional generation since the resampling step simply copies the promising sequences to replace the low-quality ones, without generating more candidate steps such as the backtracking. This ensures the potential of TSMC to be further accelerated in real-time applications. While the acceleration involves more infra optimizations to reduce the memory and inference costs, .e.g quantization, kv-cache, dynamic-batching, which are out of the scope of this work
>
> In terms of the training time, we have no data about existing methods. In our case, we generate 80 samples for each problem on 8000 problems using the parallel decoding method. The cost is around $10s*8000=80000s=22.2h$. But what we observe is that only 2000~3000 problems (25% to 40%) are enough to saturate the training of the value network.

---

> > ### Author Response · Authors · 2024-11-23
> > **Response (2/2)**
> >
> > > How does TSMC perform when applied to multi-step reasoning tasks beyond mathematical problems, such as programming or logical puzzles?
> >
> > TSMC provides a unified framework for all multi-step reasoning tasks, as long as the “step” could be well-defined.
> >
> > Following your suggestion, we included an additional experiment on the NumGLUE dataset for the natural language inference task. The reasoning process is a Python program. Please see Table 5 in Appendix G for the result. Our TSMC method still takes a consistent advantage under various numbers of voting samples. On this dataset, we separate the Python program into steps  via the newline character “\n”
> >
> > Beyond steps defined according to delimiters, TSMC can also be applied to steps defined by a fixed number of tokens (as shown above), enabling it to generalize effectively to any reasoning task. But overall, the reasoning step split is a general research problem, where different splitting strategies are expected from different domains.
> >
> >
> > > Can the method be adapted for dynamic or variable-length intermediate steps in more complex reasoning tasks?
> >
> > We would like to kindly point out that our method has already been applied to variable-length steps in our experiment. We discussed the strategy to handle the variable-length steps in Section 4.1 TSMC details, “For sequences that terminate early,we assign an incremental importance weight of 1 during the remaining resampling steps.”
> >
> > If a sequence stops early with only $t<T$ steps, then its chance of being correct has already been determined. So $V(x_{1:t+i})=V(x_{1:t})$ for $1\leq i\leq T-t$ and the incremental importance weight $V(x_{1:t+i+1})/V(x_{1:t+i})=1$.
> >
> > > How sensitive to the choice of generator and dataset size.
> >
> > We have designed our experiment to show that TSMC can be applied to different generators (Llemma & DeepSeek). In general, TSMC is especially useful for a weak generator where the generator $p$ is far from the target distribution $\sigma$, e.g, Lemma-7B on MATH500. But it is still useful when the generator is strong enough, e.g., DeepSeek-7B on GSM8K.
> >
> > Regarding the dataset size, we found that the 7B value network in TSMC could be well-trained with 2000~3000 problems. Since the only annotation needed is the ground-truth answer, the dataset of such a scale is usually not a concern in the real-world application.

---

### Author Response · Authors · 2024-11-23
**Generic Response**

We sincerely thank all the reviewers for their valuable comments and constructive suggestions.

We appreciate all the positive feedback in recognition of
- the **significance** of our research problem (UXiV, bjdp)
- the **novelty** of the proposed method (DqPL, jY8a, 16x7, bjdp)
- our comprehensive **theoretical analysis** (jY8a, 16x7)
- the promising **experimental results** (DqPL, UXiV, jY8a, bjdp)
- and the **clear writing** (jY8a, 16x7, bjdp)

In our response, we would like to highlight our effort to address the following common concerns
- We added one more easy Math benchmark Multiarith in Table 4 to show that TSMC has **no adverse effect on easy problems** (jY8a)
- We evaluated TSMC on a natural language inference benchmark NumGLUE (reasoning via Python programming) in Table 5 to demonstrate its **applicability to other reasoning tasks** (DpQL, 16x7)
- We added more discussions about the **computational cost** of TSMC (DpQL, 16x7)
- We have detailed our **selection of MATH500 problems** and presented its result on our on-hold validation set in Table 6 (jY8a, bjdp)

We have added our additional experiments to our submitted paper (Appendix G) and highlighted the change in the blue color.

---

### Comment · Area_Chair_175E · 2024-11-25

Dear Reviewers,

Thank you for your time and effort in reviewing for ICLR'25.

This is a gentle reminder to read and respond to the authors' rebuttals. Please submit your feedback in time. Thank you very much!

Best regards,

AC

---

### Meta-Review · Area_Chair_175E · 2024-12-20

**Metareview:**

This paper focuses on improving multi-step reasoning in LLMs for solving math problems. The authors introduce a new approach called Twisted Sequential Monte Carlo (TSMC) that increases sampling efficiency by concentrating on partial solutions. This method is less dependent on human input, making it more scalable. The paper shows that TSMC outperforms existing methods based on theoretical insights and experiments. Most reviewers agree that the proposed approach in this paper is novel, with a solid theoretical foundation and satisfactory experimental results. Although some reviewers expressed concerns regarding the sufficiency of the experiments and raised minor issues, the authors effectively addressed these queries, particularly those raised by reviewer bjdp and reviewer 16x7. While other responses to reviewer concerns regarding the experiments did not receive further acknowledgment from the reviewers, I believe they adequately resolved several experimental issues. Therefore, I recommend accepting this paper.

**Additional Comments On Reviewer Discussion:**

During the discussion phase, two reviewers acknowledged that the authors had addressed their concerns, with one reviewer increasing their score. Although the other reviewers did not provide feedback to the authors, they initially expressed positive opinions. Overall, I believe the authors have effectively resolved the majority of the reviewers' issues.

---

### Decision · Program_Chairs · 2025-01-22

Accept (Poster)